# Beyond Looking Up, Try Looking Around: Harmonizing Global Structure and Local Consistency in Optimal Transport for Short Text Clustering

Zhihao Yao [1]  Yuxuan Gu [2]  Jixuan Yin [1]  Bo Li [1]

## Abstract

Pseudo-labeling based on Optimal Transport (OT) has become an effective mechanism for enhancing short text clustering. Existing OT methods are short in modeling semantic consistencies between samples, which may assign different pseudo-labels to semantically similar samples. These erroneous pseudo-labels can cause the model to produce inferior clusters. This paper proposes a novel short text clustering framework, which remedies the neglect of semantic consistency in existing OT methods, generating reliable pseudo-labels to facilitate clustering. Specifically, the proposed approach first designs an instance-level attention mechanism to capture semantic relationships between samples, which are then integrated into the OT formulation to endow the transport process with neighborhood semantic awareness. By solving the proposed OT formulation, reliable pseudo-labels are obtained that simultaneously account for sample-to-sample semantic consistency and sample-to-cluster global structure information. These pseudo-labels are then used as supervisory signals to guide the model to achieve accurate clustering. Extensive experiments demonstrate that the proposed approach outperforms state-of-the-art methods. The code is available at: https://github.com/YZH0905/CAOT-STC.

## 1. Introduction

In the era of massive digital communication, short texts such as messages, queries, and comments are generated on an unprecedented scale. Effectively clustering and organizing these short texts is crucial for numerous applications

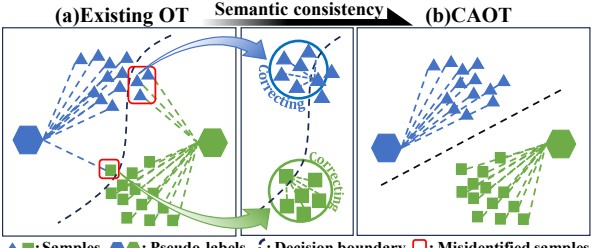

*Figure 1.* Schematic illustration of the motivation. Existing methods rely solely on the transport cost from samples to candidate pseudo-labels (hexagons), where the cost is visualized as the distances from triangles to hexagons, causing semantically similar neighbors to be assigned different pseudo-labels (red boxes). The proposed CAOT addresses this issue by incorporating sample-to-sample semantic consistency, thereby correcting these misassignments.

(Ma & Zheng, 2025). This urgency is further amplified by the prevalence of Large Language Models (LLMs), since LLM-driven applications, such as chatbots (Kang & Ki, 2025) and virtual assistants (Kweon et al., 2025), have further fueled the proliferation of short texts. Existing methods typically adopt an ***EM-like optimization framework*** to achieve clustering (Zheng et al., 2023; Wu et al., 2025), which alternates between estimating pseudo-labels (E-step) and updating model parameters based on the generated pseudo-labels (M-step). Under this framework, early methods often adopt a greedy strategy to estimate pseudo-labels directly from model predictions (Sohn et al., 2020). However, they ignore both global and local information of the data distribution (Qin et al., 2026), resulting in unreliable pseudo-labels that may misguide model optimization in the subsequent M-step (Yin et al., 2025; Cui et al., 2025).

Instead of assigning pseudo-labels greedily, recent studies introduce Optimal Transport (OT) to estimate pseudo-labels by minimizing the overall cost of transporting the sample distribution to the cluster distribution (Zheng et al., 2023; Cui et al., 2025), integrating sample-cluster relationships from a global perspective (Zhang et al., 2024; Zheng et al., 2023), i.e., the ***sample-to-cluster global structure***. However, there is still a critical limitation: they ignore predictive consistency between semantically similar samples, i.e., ***sample-to-sample semantic consistency***. Relying only on minimizing overall transport cost, these methods may assign differ-

---

[1]Harbin Engineering University, China. [2]Harbin Institute of Technology, China. Correspondence to: Bo Li <boli@hrbeu.edu.cn>.

*Proceedings of the 43rd International Conference on Machine Learning*, Seoul, South Korea. PMLR 306, 2026. Copyright 2026 by the author(s).

ent pseudo-labels to semantically similar samples when the transport costs to candidate pseudo-labels are not sufficiently distinct or accurate. As illustrated in Figure 1(a), the blue triangles inside the red box exhibit nearly identical transport costs to both hexagons. Due to this ambiguity, existing OT methods struggle to make correct estimation, potentially assigning these samples to erroneous green pseudo-labels, even though they are surrounded by a dense neighborhood of correct blue peers. This issue is common in practice because feature spaces derived from pretrained encoders often lack clear category boundaries (Zhang et al., 2021), causing samples near boundaries to have similar transport costs to different candidate pseudo-labels. Once these noisy pseudo-labels are introduced into the M-step, the initial embedding biases are amplified, leading to widespread mislabeling and significantly degrading clustering quality.

To address this limitation, we propose a novel OT method to enhance pseudo-label estimation in the E-step, termed *consistency-aware adaptive optimal transport* (**CAOT**). In contrast to existing OT methods, CAOT incorporates the semantic relationships among surrounding samples into the transport process, correcting erroneous estimations caused by ambiguous transport costs, as shown in Figure 1(b). To achieve this, we first propose a novel instance-level attention network to capture semantic relationships between samples. These learned relationships are integrated into CAOT as a semantic consistency constraint, encouraging semantically similar samples to be assigned identical pseudo-labels. During EM iterations, the instance-level attention network and the clustering assignments are jointly optimized in the M-step, using pseudo-labels inferred by CAOT (E-step) as supervisory signals. Through successive EM iterations, the attention network becomes increasingly effective in modeling semantic relationships, which in turn provides accurate guidance for CAOT to promote semantic consistency. As a result, the pseudo-labels generated by CAOT capture both the sample-to-cluster global structure and the sample-to-sample semantic consistency. These reliable pseudo-labels can effectively optimize model parameters to achieve satisfactory clustering performance.

We conduct comprehensive experiments on eight benchmark datasets. Experimental results demonstrate that the proposed approach outperforms state-of-the-art methods. Complementing these quantitative results, visualization comparisons of transport results further demonstrate that pseudo-labels produced by the proposed CAOT are more accurate and confident than those produced by existing OT methods. Beyond the field of short-text, the proposed approach can be generalized to long-text and image domains, demonstrating its potential as a general solution for various clustering tasks. Further analysis demonstrates that the proposed approach also has computational advantages over existing methods, which makes it suitable for real-world clustering tasks.

## 2. Preliminary

This section reviews the commonly adopted OT formulation and its application in pseudo-labeling. Given a source distribution $\boldsymbol{a}$ and a target distribution $\boldsymbol{b}$, the objective of OT is to find a matrix $\boldsymbol{Q}$ to transform the distribution $\boldsymbol{a}$ to $\boldsymbol{b}$ with minimal total cost. The optimization problem in OT is:

$$\min_{\boldsymbol{Q}} \ \langle \boldsymbol{Q}, \boldsymbol{C} \rangle + \varepsilon_1 \langle \boldsymbol{Q}, \log(\boldsymbol{Q}) - \mathbf{1}_{|\boldsymbol{a}| \times |\boldsymbol{b}|} \rangle$$
$$\text{s.t. } \boldsymbol{Q}\mathbf{1}_{|\boldsymbol{b}| \times 1} = \boldsymbol{a}, \ \boldsymbol{Q}^T \mathbf{1}_{|\boldsymbol{a}| \times 1} = \boldsymbol{b}, \quad (1)$$

where $\langle \cdot, \cdot \rangle$ represents the Frobenius inner product, $\varepsilon_1$ is a weighting coefficient, $\boldsymbol{C} \in \mathbb{R}^{|\boldsymbol{a}| \times |\boldsymbol{b}|}$ is a cost matrix, and $|\boldsymbol{a}|$ and $|\boldsymbol{b}|$ are the dimensions of $\boldsymbol{a}$ and $\boldsymbol{b}$, respectively.

In OT-based pseudo-labeling methods, the matrix $\boldsymbol{C}$ is defined as $-\log(\boldsymbol{P})$, where $\boldsymbol{P} \in \mathbb{R}^{n \times k}$ is a matrix containing the predicted probabilities of samples, with $n$ the batch size and $k$ the number of categories. The optimization problem is:

$$\min_{\boldsymbol{Q}} \ \langle \boldsymbol{Q}, -\log(\boldsymbol{P}) \rangle + \varepsilon_1 \langle \boldsymbol{Q}, \log(\boldsymbol{Q}) - \mathbf{1}_{n \times k} \rangle$$
$$\text{s.t. } \boldsymbol{Q}\mathbf{1}_{k \times 1} = \frac{1}{n}\mathbf{1}_{n \times 1}, \ \boldsymbol{Q}^T \mathbf{1}_{n \times 1} = \frac{1}{k}\mathbf{1}_{k \times 1}, \quad (2)$$

where $\boldsymbol{Q}\mathbf{1}_{k \times 1} = \frac{1}{n}\mathbf{1}_{n \times 1}$ indicates that all samples are equally important, and $\boldsymbol{Q}^T \mathbf{1}_{n \times 1} = \frac{1}{k}\mathbf{1}_{k \times 1}$ ensures that each category in the target distribution is assigned the same probability. After obtaining the transport matrix $\boldsymbol{Q}$, the pseudo-label for the $i$-th sample is calculated as $\hat{y}_i = \arg\max(\boldsymbol{Q}_{i:})$.

## 3. The Proposed Method

This section provides a detailed description of the proposed approach. §3.1 gives a high-level overview of the entire framework. §3.2, 3.3, and 3.4 offer an in-depth description of three sub-components, where the proposed CAOT is introduced in §3.2 to address the limitation of existing OT methods. §3.5 introduces the training pipeline.

### 3.1. The Overall Structure

The structure of the proposed model is shown in Figure 2. It consists of three components: the Pseudo-label Generation Module (PGM), the Semantic Similarity Construction Module (SSCM), and the Supervised Guidance Module (SGM).

During model training, the PGM functions in the E-step, where it performs CAOT using the similarity matrix $\boldsymbol{S}^{att}$ to generate pseudo-labels $\hat{\boldsymbol{y}}$. The SSCM and SGM operate in the M-step to update model parameters. The role of SSCM is to capture the semantic similarities between samples and construct the similarity matrix $\boldsymbol{S}^{att}$. To this end, contrastive learning is first applied to enhance the discriminability of sample representations ($L_I$ loss), after which pseudo-labels $\hat{\boldsymbol{y}}$ are leveraged to train the instance-level attention network $G_h$ to yield the matrix $\boldsymbol{S}^{att}$ ($L_A$ loss). The role of SGM is to leverage pseudo-labels $\hat{\boldsymbol{y}}$ to optimize cluster assignments.

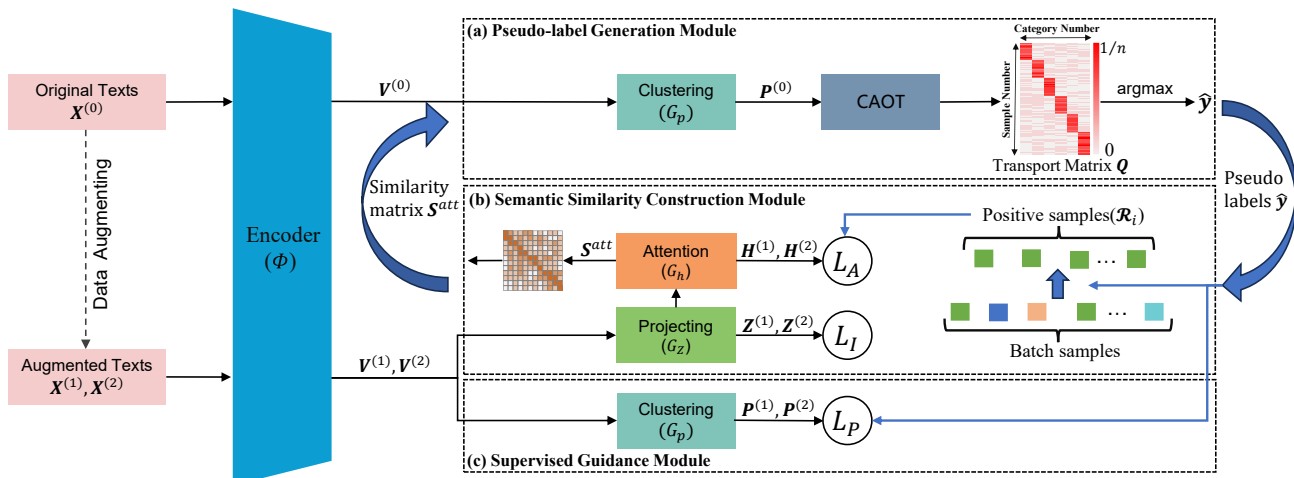

*Figure 2.* The overall architecture of the proposed method. It comprises three components: (a) Pseudo-label Generation Module, (b) Semantic Similarity Construction Module, and (c) Supervised Guidance Module.

## 3.2. The Pseudo-label Generation Module

The Pseudo-label Generation Module is expected to produce reliable pseudo-labels. To this end, we propose a novel OT formulation, referred to as CAOT. Solving the optimization problem in CAOT yields pseudo-labels that simultaneously integrate sample-to-cluster global structural information and sample-to-sample semantic consistency.

The structure of the Pseudo-label Generation Module is shown in Figure 2(a). Given a batch of $n$ text samples $X^{(0)} = [x_1^{(0)}; \ldots; x_n^{(0)}]$, the text representations are obtained by the Encoder $\Phi$ as $V^{(0)} \in \mathbb{R}^{n \times d_1}$, where $d_1$ is the dimension of the representations. Then, the Clustering Network $G_p$ is used to predict probability assignments $P^{(0)} \in \mathbb{R}^{n \times k}$ ($G_p$ is an MLP network with dimensions 768-768-768-$k$). After that, pseudo-labels can be generated by solving the optimization problem in CAOT, which is defined as follows:

$$\min_{Q,b} \langle Q, -\log(P^{(0)}) \rangle + \varepsilon_1 H(Q) + \varepsilon_2 (\Psi(b)^T \mathbf{1}_{k \times 1}) - \varepsilon_3 \langle S, QQ^T \rangle$$

$$\text{s.t. } Q\mathbf{1}_{k \times 1} = a, \ Q^T\mathbf{1}_{n \times 1} = b, \ b^T\mathbf{1}_{k \times 1} = 1, \tag{3}$$

where $\varepsilon_1$, $\varepsilon_2$ and $\varepsilon_3$ are weighting coefficients, $a = \frac{1}{n}\mathbf{1}_{n \times 1}$, and $b$ remains to be determined. $S = S^{cos} + S^{att}$ is the semantic similarity matrix, where $S^{cos}$ is the cosine similarity calculated from $P^{(0)}$, and $S^{att}$ is the attention similarity generated from the instance-level attention network ($G_h$) via Eq. (8). Each term in Eq. (3) is explained as follows:

- $H(Q) = \langle Q, \log(Q) - \mathbf{1}_{n \times k} \rangle$ is the same as Eq. (2), which prevents $Q$ from becoming too sparse while ensuring that all elements remain strictly non-negative.

- $\Psi(b) = -\log(b) - \log(\mathbf{1}_{k \times 1} - b)$. Unlike Eq. (2), which strictly enforces the hard constraint $Q^T\mathbf{1}_{n \times 1} = \frac{1}{k}\mathbf{1}_{k \times 1}$, CAOT adopts $\Psi(b)^T\mathbf{1}_{k \times 1}$ as an adjustable penalty item to

encourage $Q^T\mathbf{1}_{n \times 1}$ to approach a uniform distribution. By adjusting the strength of this item, CAOT can flexibly adapt to datasets with varying degrees of category imbalance.

- $\langle S, QQ^T \rangle$ incorporates semantic consistency into CAOT, encouraging the transport matrix $Q$ to align with the intrinsic semantic relationships between samples. Specifically, this design encourages the transport vector $Q_{i:}$ to be similar to $Q_{j:}$ when $S_{ij}$ is large. In other words, CAOT drives semantically similar samples towards similar transport vectors, thereby producing the same pseudo-labels.

In the proposed CAOT, $\langle Q, -\log(P^{(0)}) \rangle$ incorporates the *sample-to-cluster global structure* into $Q$ through the global minimum-cost matching constraint, enabling CAOT to exploit the joint confidence over the entire batch to correct erroneous predictions for some samples in $P^{(0)}$. Simultaneously, $\langle S, QQ^T \rangle$ integrates *sample-to-sample semantic consistency* in $Q$, ensuring that semantically similar samples are assigned the same pseudo-labels. These two properties in $Q$ empower the generation of reliable pseudo-labels.

To solve the proposed optimization problem in CAOT, we design an iterative method that integrates Taylor expansion and Lagrange multiplier algorithm. The detailed solution is provided in Appendix A.1, and the convergence analysis of the proposed solution is given in Appendix A.2. Once the transport matrix $Q$ is available, the pseudo-label for the $i$-th sample can be generated as $\hat{y}_i = \arg\max_j Q_{ij}$. The resulting pseudo-labels are then utilized in SSCM and SGM to update the model parameters during the M-step.

## 3.3. Semantic Similarity Construction Module

The Semantic Similarity Construction Module is designed to identify and quantify semantic similarities between samples. The resulting similarity information is integrated into

Eq. (3), guiding CAOT to incorporate sample-to-sample semantic consistency into the E-step.

The structure of the Semantic Similarity Construction Module is shown in Figure 2(b). Inspired by Socrates' saying "To know others, first know yourself.", we first enhance the discriminability of each sample representation using contrastive learning, providing a foundation for similarity modeling. Given a batch of texts $X^{(0)}$, *contextual augmenter* technique (Kobayashi, 2018) is used to generate two augmented texts, denoted as $X^{(1)}$ and $X^{(2)}$. The representations of the augmented texts are then obtained by the Encoder $\Phi$ as $\boldsymbol{V}^{(1)} \in \mathbb{R}^{n \times d_1}$ and $\boldsymbol{V}^{(2)} \in \mathbb{R}^{n \times d_1}$, respectively. Subsequently, the Projecting Network $G_z$ is employed to reduce the dimensionality of the representations ($G_z$ is an MLP network with dimensions 768-768-128). The outputs are $\boldsymbol{Z}^{(1)} \in \mathbb{R}^{n \times d_2}$ and $\boldsymbol{Z}^{(2)} \in \mathbb{R}^{n \times d_2}$, respectively.

Then, contrastive learning improves the discriminability of each sample representation by pulling positive pairs together and pushing negative pairs apart. Let $\boldsymbol{z}_i^{(1)}$ and $\boldsymbol{z}_i^{(2)}$ be the $i$-th rows in $\boldsymbol{Z}^{(1)}$ and $\boldsymbol{Z}^{(2)}$, respectively. For $\boldsymbol{z}_i^{(1)}$, its corresponding positive pair is $\{\boldsymbol{z}_i^{(1)}, \boldsymbol{z}_i^{(2)}\}$, while the negative pairs are $\{\boldsymbol{z}_i^{(1)}, \boldsymbol{z}_i^{(1)}\}$ and $\{\boldsymbol{z}_i^{(1)}, \boldsymbol{z}_i^{(2)}\}$ for all $v \neq i$. For $\boldsymbol{z}_i^{(2)}$, the positive and negative pairs are constructed in the same manner. The loss function of contrastive learning for the $i$-th sample is defined as follows:

$$l_{I,i} = -\log \frac{e^{sim(\boldsymbol{z}_i^{(1)}, \boldsymbol{z}_i^{(2)})/\tau_I}}{\sum_{v=1, v \neq i}^n (e^{sim(\boldsymbol{z}_i^{(1)}, \boldsymbol{z}_v^{(1)})/\tau_I} + e^{sim(\boldsymbol{z}_i^{(1)}, \boldsymbol{z}_v^{(2)})/\tau_I})}$$
$$- \log \frac{e^{sim(\boldsymbol{z}_i^{(2)}, \boldsymbol{z}_i^{(1)})/\tau_I}}{\sum_{v=1, v \neq i}^n (e^{sim(\boldsymbol{z}_i^{(2)}, \boldsymbol{z}_v^{(1)})/\tau_I} + e^{sim(\boldsymbol{z}_i^{(2)}, \boldsymbol{z}_v^{(2)})/\tau_I})}, \quad (4)$$

where $sim(\boldsymbol{z}_i^{(1)}, \boldsymbol{z}_v^{(2)})$ is the cosine similarity between two representations, and $\tau_I$ is the temperature parameter. The final loss function of contrastive learning is calculated over a batch and defined as $L_I = \frac{1}{2n} \sum_{i=1}^n l_{I,i}$.

By treating all samples with $v \neq i$ as negative pairs, contrastive learning encourages sample representations to be uniformly distributed in the feature space. Thus, a highly discriminative representation can be learned for each sample. This design facilitates the modeling of semantic similarity between samples, as discriminative representations possess the necessary clarity to distinguish each other (Zhang et al., 2021). Based on this, we propose an instance-level attention network to explore similarity relationships between samples. The structure of this attention network $G_h$ is shown in Figure 3. It takes $\boldsymbol{Z}^{(1)}$ and $\boldsymbol{Z}^{(2)}$ as inputs. For $\boldsymbol{Z}^{(1)}$, the network performs three linear transformations:

$$\boldsymbol{K}_1^{(1)} = \boldsymbol{Z}^{(1)} \boldsymbol{W}_{K_1}, \ \boldsymbol{K}_2^{(1)} = \boldsymbol{Z}^{(1)} \boldsymbol{W}_{K_2}, \ \boldsymbol{T}^{(1)} = \boldsymbol{Z}^{(1)} \boldsymbol{W}_T, \quad (5)$$

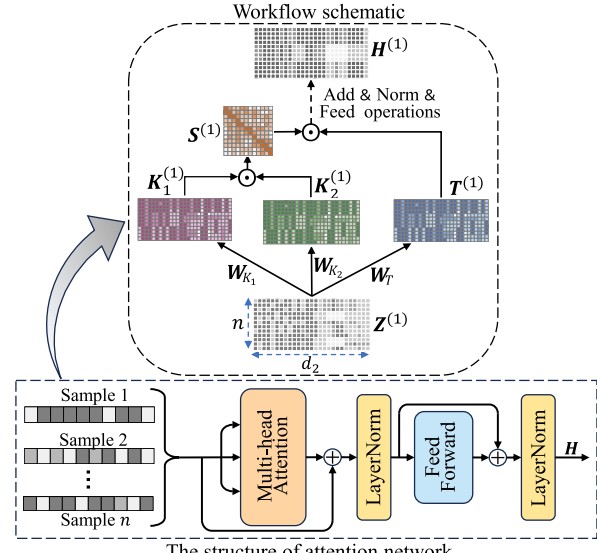

*Figure 3.* The instance-level attention network.

after which the similarity matrix $\boldsymbol{S}^{(1)}$ is computed:

$$\boldsymbol{S}^{(1)} = \text{Softmax} \left( (\boldsymbol{K}_1^{(1)} \boldsymbol{K}_2^{(1)T}) / \sqrt{d_2} \right). \quad (6)$$

Finally, the rows of $\boldsymbol{T}^{(1)}$ are linearly combined according to the similarity matrix $\boldsymbol{S}^{(1)}$ as follows:

$$\boldsymbol{h}_i^{(1)} = \sum_{j=1}^n S_{ij}^{(1)} \boldsymbol{t}_j^{(1)}, \ \ \boldsymbol{H}^{(1)} = [\boldsymbol{h}_1^{(1)}; \boldsymbol{h}_2^{(1)}; \ldots; \boldsymbol{h}_n^{(1)}], \quad (7)$$

where $\boldsymbol{t}_j^{(1)}$ is the $j$-th row of $\boldsymbol{T}^{(1)}$, and $S_{ij}^{(1)}$ is the $(i,j)$-th element of $\boldsymbol{S}^{(1)}$. Similarly, $\boldsymbol{S}^{(2)}$ and $\boldsymbol{H}^{(2)}$ are obtained by passing $\boldsymbol{Z}^{(2)}$ through the Attention Network. The attention similarity matrix is calculated as follows:

$$\boldsymbol{S}^{att} = (\boldsymbol{S}^{(1)} + \boldsymbol{S}^{(2)})/2. \quad (8)$$

To ensure that the similarity matrix $\boldsymbol{S}^{att}$ can accurately reflects the underlying semantic relationships between samples, we apply targeted training to $G_h$. Let $\boldsymbol{H} = [\boldsymbol{H}^{(1)}, \boldsymbol{H}^{(2)}] \in \mathbb{R}^{2n \times d_2}$. As indicated in Eq. (7), $\boldsymbol{S}^{att}$ is intrinsically linked to $\boldsymbol{H}$. Based on this, we optimize $\boldsymbol{S}^{att}$ by applying targeted training to $\boldsymbol{H}$ as a proxy. We achieve this by enhancing the intra-cluster compactness and inter-cluster separation of $\boldsymbol{H}$, which implicitly guides $\boldsymbol{S}^{att}$ to assign high attention weights to samples belonging to the same cluster. In practice, we first use pseudo-labels to identify samples belonging to the same cluster. For the $i$-th sample, the set of indices belonging to the same cluster is defined as follows:

$$\mathcal{R}_i = \{j \mid \hat{y}_j = \hat{y}_i, j = 1, ..., n\}, \quad (9)$$

where $\hat{y}_i$ is the pseudo-label of the $i$-th sample. After that,

the loss for the $i$-th sample is defined as follows:

$$l_{A,i} = -\log\frac{\sum_{j\in\mathcal{R}_i}S_{ij}^{att}[e^{\text{sim}(\boldsymbol{h}_i,\boldsymbol{h}_j)/\tau_A} + e^{\text{sim}(\boldsymbol{h}_i,\boldsymbol{h}_{n+j})/\tau_A}]}{\sum_{v=1,v\neq i,v\neq n+i}^{2n}e^{\text{sim}(\boldsymbol{h}_i,\boldsymbol{h}_v)/\tau_A}}$$
$$- \log\frac{\sum_{j\in\mathcal{R}_i}S_{ij}^{att}[e^{\text{sim}(\boldsymbol{h}_{n+i},\boldsymbol{h}_j)/\tau_A} + e^{\text{sim}(\boldsymbol{h}_{n+i},\boldsymbol{h}_{n+j})/\tau_A}]}{\sum_{v=1,v\neq i,v\neq n+i}^{2n}e^{\text{sim}(\boldsymbol{h}_{n+i},\boldsymbol{h}_v)/\tau_A}},$$
(10)

where $S_{ij}^{att}$ is the $(i,j)$-th element of $\boldsymbol{S}^{att}$, $\boldsymbol{h}_i$ is the $i$-th row of $\boldsymbol{H}$, and $\tau_A$ is the temperature parameter. The final loss function of attention optimization is calculated over a batch and defined as $L_A = \frac{1}{2n}\sum_{i=1}^{n}l_{A,i}$.

By minimizing $L_A$, samples within the same cluster are assigned high weights in $\boldsymbol{S}^{att}$, which in turn facilitates CAOT in the subsequent E-step. In the training process, accurate pseudo-labels help establish an accurate $\boldsymbol{S}^{att}$ (M-step), which in turn facilitates the generation of more reliable pseudo-labels than those in the previous iteration (E-step).

### 3.4. Supervised Guidance Module

The Supervised Guidance Module leverages pseudo-labels generated by CAOT as supervision targets to optimize the model in clustering. The structure of the Supervised Guidance Module is shown in Figure 2(c). Given a batch of representations $\boldsymbol{V}^{(1)}$ and $\boldsymbol{V}^{(2)}$, the corresponding probability assignments $\boldsymbol{P}^{(1)}$ and $\boldsymbol{P}^{(2)}$ are obtained. Then, following the paradigm of supervised classification, the model is optimized by aligning its predictive probabilities with the supervision targets. The loss function is defined as follows:

$$L_P = -\frac{1}{n}\sum_{i=1}^{n}(\text{OneHot}(\hat{y}_i)\log\boldsymbol{p}_i^{(1)} + \text{OneHot}(\hat{y}_i)\log\boldsymbol{p}_i^{(2)}),$$
(11)

where $\text{OneHot}(\cdot)$ is a one-hot encoding operator, $\hat{y}_i$ is the pseudo-label for the $i$-th sample, $\boldsymbol{p}_i^{(1)}$ and $\boldsymbol{p}_i^{(2)}$ are the $i$-th rows of the probability matrix $\boldsymbol{P}^{(1)}$ and $\boldsymbol{P}^{(2)}$, respectively.

### 3.5. Training Procedure

To prevent error accumulation in EM iterative optimization, we introduce a warm-up stage to provide a stable starting point for the training process. In this stage, pseudo-labels are generated via K-means on the representations $\boldsymbol{V}^{(0)}$, rather than the proposed CAOT. The warm-up loss combines $L_I$ and $L_A$, defined as follows:

$$L = L_A + \lambda L_I,$$
(12)

where $\lambda$ is a balancing hyperparameter. The *stop gradient* technique is applied to the backpropagation of $L_A$, ensuring that $L_A$ only optimizes the Attention Network $G_h$. After the warm-up stage, CAOT is utilized to generate pseudo-labels. The overall loss at this stage is formulated as follows:

$$L = L_A + L_P + \lambda L_I.$$
(13)

---

**Algorithm 1** The overall training procedure

**Input:** dataset $\mathcal{X}^{(0)}$; warm-up epochs $E_{warm}$; total epochs $E_{total}$; Encoder $\Phi$; Network $G_z$, $G_h$, and $G_p$.
**Output:** model parameters.
**Procedure:**
1:  $\mathcal{X}^{(2)}, \mathcal{X}^{(1)} \leftarrow \text{Augmenter}(\mathcal{X}^{(0)})$ ▷ Generate augmented dataset
2:  **for** $epoch = 1$ **to** $E_{total}$ **do**
3:    $\boldsymbol{X}^{(0)}, \boldsymbol{X}^{(1)}, \boldsymbol{X}^{(2)} \sim \mathcal{X}^{(0)}, \mathcal{X}^{(1)}, \mathcal{X}^{(2)}$   ▷ Sample batches
4:    Compute representations $\boldsymbol{V}^{(0)}, \boldsymbol{Z}^{(1)}, \boldsymbol{Z}^{(2)}, \boldsymbol{H}^{(1)}, \boldsymbol{H}^{(2)}$ and probability assignments $\boldsymbol{P}^{(1)}, \boldsymbol{P}^{(2)}$.
5:    **if** $epoch \leq E_{warm}$ **then**
6:      **Step 1:** warm-up stage
7:      $\hat{\boldsymbol{y}} \leftarrow \text{K-means}(\boldsymbol{V}^{(0)})$         ▷ Compute pseudo-labels
8:      $\mathcal{R} \leftarrow \hat{\boldsymbol{y}}$       ▷ Identify same-cluster samples by Eq.(9)
9:      $L \leftarrow L_A + \lambda L_I$       ▷ Calculate the loss in Eq.(12)
10:     Update parameters of $\Phi$, $G_z$ and $G_h$.
11:   **else**
12:     **Step 2:** interactive optimization stage
13:     $\boldsymbol{S}^{att} \leftarrow G_h(\boldsymbol{Z}^{(1)}, \boldsymbol{Z}^{(2)})$ ▷ Compute the attention similarity
14:     $\hat{\boldsymbol{y}} \leftarrow \text{CAOT}(\boldsymbol{P}^{(0)}, \boldsymbol{S}^{att})$         ▷ Compute pseudo-labels
15:     $\mathcal{R} \leftarrow \hat{\boldsymbol{y}}$       ▷ Identify same-cluster samples by Eq.(9)
16:     $L \leftarrow L_A + L_P + \lambda L_I$     ▷ Calculate the loss in Eq. (12)
17:     Update parameters of $\Phi$, $G_z$, $G_h$, and $G_p$.
18:   **end if**
19: **end for**

---

After training, for a given text $x^{(0)}$, its clustering result is given by $\arg\max(\boldsymbol{p}^{(0)})$, where $\boldsymbol{p}^{(0)}$ is the output of the Clustering Network $G_p$. The pseudo-code of the overall training process is provided in Algorithm 1.

## 4. Experiments

In this section, we present systematic evaluation experiments. §4.1 introduces the configurations. §4.2 analyzes the performance of clustering. § 4.3 compares the quality of feature spaces learned by the proposed model and the baseline methods. §4.4 presents ablation studies to evaluate the contribution of each component of the model. §4.5 conducts sensitivity analysis to examine the stability of the model under different hyperparameter settings.

### 4.1. Experiment Setups

**Datasets.**  We conduct experiments on eight benchmark datasets: AgNews, StackOverflow, Biomedical, Search-Snippets, GoogleNews-TS, GoogleNews-T, GoogleNews-S, and Tweet. Table 1 summarizes the characteristics of each dataset. Depending on the degree of category imbalance (R), AgNews, StackOverflow, and Biomedical are regarded as balanced datasets; SearchSnippets as a slightly imbalanced

dataset; GoogleNews-TS, GoogleNews-T, and GoogleNews-S as imbalanced datasets; Tweet as a severely imbalanced dataset. Detailed information about these datasets is provided in Appendix C.1.

**Experiment Settings.** We use distilbert-base-nli-stsb-mean-tokens (SBERT) from the Sentence Transformers library as the Encoder (Reimers & Gurevych, 2019). All parameters are optimized using the Adam optimizer. The learning rate of the Encoder is $5 \times 10^{-6}$, and that of other networks is $5 \times 10^{-4}$. The maximum input length for the Encoder is 32. The output dimensions of the Encoder and the Projection Network are set to $d_1 = 768$ and $d_2 = 128$, respectively. We adopt Accuracy (ACC) and Normalized Mutual Information (NMI) to evaluate the model. The definitions of the evaluation metrics and other settings are provided in Appendix C.2 and Appendix C.3, respectively.

**Baselines.** We compare the proposed method with the following baselines: **TF-IDF** (Chowdhury, 2010) extracts text representations using the TF-IDF technique. **Back-Trans** (Shen et al., 2019) generates representations via contrastive learning with back-translation. **STCC** (Xu et al., 2017) uses a CNN to optimize the output of word2vec as representations. **Self-Train** (Hadifar et al., 2019) learns representations with an auto-encoder and updates them according to cluster assignments. **SimCSE** (Gao et al., 2021) generates representations through contrastive learning with Dropout. For all the above methods, K-means is applied on the learned representations to yield clustering results. **ESimCSE** (Wu et al., 2022) extends SimCSE by addressing the influence of sentence length on positional embeddings in transformers. **SCCL** (Zhang et al., 2021) employs contrastive learning to refine the output of SBERT as representations and obtains clustering results using the DEC algorithm (Xie et al., 2016). **RSTC** (Zheng et al., 2023) constructs pseudo-labels using optimal transport to assist clustering. **SCL** (Yong et al., 2024) learns a projection matrix to map samples into a contrastive subspace, aligning original and projected embeddings to improve clustering. **SCPCL** (Liu et al., 2025) employs optimized adaptive optimal transport to generate pseudo-labels, which guide a contrastive learning module to learn robust representations. **FNSCC** (Zheng et al., 2025) enhances contrastive clustering by integrating fuzzy neighborhood information to refine both instance representations and soft cluster assignments.

### 4.2. Main Results

The results are presented in Table 2. The results for Back-Trans, SimCSE, ESimCSE, and SCL are from the SCL paper. RSTC, SCPCL, and FNSCC are reproduced using the configurations provided in corresponding original papers, and the remaining baseline results are from the RSTC paper. According to the results, one can see that: (1) Conventional methods, including **TF-IDF** and **Back-Trans**, per-

*Table 1.* Key information of datasets. "S" represent the dataset size; "C" is the number of categories; "L" is the average sentence length; "R" is the size ratio of the largest to the smallest category.

| Datasets | S | C | L | R |
|---|---|---|---|---|
| AgNews | 8000 | 4 | 23 | 1 |
| SearchSnippets | 12340 | 8 | 18 | 7 |
| StackOverflow | 20000 | 20 | 8 | 1 |
| Biomedical | 20000 | 20 | 13 | 1 |
| GoogleNews-TS | 11109 | 152 | 28 | 143 |
| GoogleNews-T | 11109 | 152 | 6 | 143 |
| GoogleNews-S | 11109 | 152 | 22 | 143 |
| Tweet | 2472 | 89 | 9 | 249 |

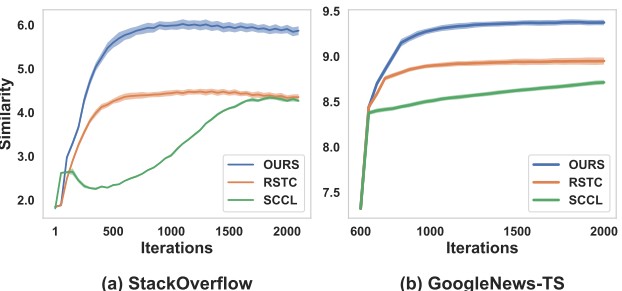

*Figure 4.* Comparison of representations. Shaded regions indicate the variance derived from 50 runs with different seeds.

form poorly due to their inability to produce meaningful representations. (2) Deep neural network-based methods (**STCC**, **Self-Train**, **SimCSE**, **ESimCSE**, and **SCL**) outperform conventional methods in learning effective representations. However, the decoupled feature learning and clustering processes lead to suboptimal results. (3) Deep joint clustering methods (**SCCL**, **RSTC**, **SCPCL**, and **FNSCC**) improve the clustering performance significantly by adopting contrastive learning to fine-tune pre-trained models. However, they ignore the relationships between samples, resulting in suboptimal performance. (4) The results of **the proposed approach** ranks first in eleven indicators and second in the remaining five indicators. Notably, on the StackOverflow dataset, the proposed method achieves a substantial accuracy gain of **5.01%**. In comparison, existing methods often perform well on some datasets but exhibit performance degradation on others, indicating limited generalization across diverse datasets. These results validate the effectiveness and generalizability of the proposed approach.

### 4.3. Comparison of representation quality

This section investigates the efficacy of the proposed approach by examining the quality of the learned feature space. We evaluate the proposed approach along with SCCL and RSTC on the SearchSnippets and GoogleNews-TS datasets. The normalized average cosine similarity of samples within

*Table 2.* Clustering results on benchmarks. Here, $\Delta$ denotes the improvement; **Bold** and underline indicate the best and second performance.

| Method | AgNews | | SearchSnippets | | Stackoverflow | | Biomedical | |
|---|---|---|---|---|---|---|---|---|
| | ACC | NMI | ACC | NMI | ACC | NMI | ACC | NMI |
| TF-IDF | 34.39 | 12.19 | 30.85 | 18.67 | 58.52 | 59.02 | 29.13 | 25.12 |
| Back-Trans | 82.70 | 57.50 | 78.30 | 64.80 | 77.30 | **75.50** | - | - |
| STCC | - | - | 76.98 | 62.56 | 51.14 | 49.10 | 43.37 | 38.02 |
| Self-Train | - | - | 72.69 | 56.74 | 59.38 | 52.81 | 40.06 | 34.46 |
| SimCSE | 81.60 | 55.50 | 74.00 | 58.50 | 75.20 | 73.60 | - | - |
| ESimCSE | 82.90 | 57.80 | 73.40 | 61.20 | 76.20 | 74.90 | - | - |
| SCCL | 83.10 | 61.96 | 79.90 | 63.78 | 70.83 | 69.21 | 42.49 | **39.16** |
| RSTC | 85.99 | 64.14 | 79.83 | **68.76** | 80.07 | 72.28 | 45.69 | 38.57 |
| SCL | 84.10 | 63.30 | 78.90 | 66.20 | 77.80 | 75.40 | - | - |
| SCPCL | 85.71 | 64.51 | 79.04 | 67.27 | 80.95 | 72.12 | 44.27 | 38.17 |
| FNSCC | 87.05 | 65.49 | **85.16** | 67.02 | 70.87 | 71.69 | 35.54 | 33.36 |
| OURS | **87.30** | **66.55** | 80.12 | 67.40 | **85.96** | 75.43 | **47.37** | 39.12 |
| $\Delta$ | +0.25 | +1.06 | -5.04 | -1.36 | +5.01 | -0.07 | +1.68 | -0.04 |

| Method | GoogleNews-TS | | GoogleNews-T | | GoogleNews-S | | Tweet | |
|---|---|---|---|---|---|---|---|---|
| | ACC | NMI | ACC | NMI | ACC | NMI | ACC | NMI |
| TF-IDF | 69.00 | 87.78 | 58.36 | 79.14 | 62.30 | 83.00 | 54.34 | 78.47 |
| Back-Trans | 72.20 | 89.80 | 60.80 | 80.70 | 67.10 | 85.00 | 59.70 | 82.30 |
| SimCSE | 71.40 | 89.10 | 60.10 | 80.30 | 63.80 | 83.60 | 61.10 | 82.90 |
| ESimCSE | 72.00 | 89.80 | 62.70 | 82.00 | 65.20 | 84.90 | 57.90 | 82.10 |
| SCCL | 82.51 | 93.01 | 69.01 | 85.10 | 73.44 | 87.98 | 73.10 | 86.66 |
| RSTC | 83.30 | 92.62 | 73.10 | 87.47 | 78.11 | 89.01 | 77.75 | 86.07 |
| SCL | 74.30 | 89.60 | 63.20 | 82.20 | 68.80 | 85.10 | 64.20 | 84.10 |
| SCPCL | **86.97** | 92.08 | 72.95 | 85.83 | 79.06 | 88.35 | 80.34 | 86.64 |
| FNSCC | 81.04 | 92.17 | 68.61 | 84.43 | 71.73 | 85.72 | 81.65 | 87.17 |
| OURS | 83.53 | **93.15** | **73.47** | **87.54** | **79.57** | **89.30** | **82.36** | **89.49** |
| $\Delta$ | -3.44 | +0.14 | +0.37 | +0.07 | +0.71 | +0.29 | +0.65 | +2.32 |

the same category is used as the evaluation metric, which is defined as follows:

$$NS = \frac{\sum_{i=1}^{n} \sum_{j=1}^{n} \mathbb{1}(y_i = y_j) \times sim(z_i^{(0)}, z_i^{(1)})}{\sum_{i=1}^{n} \sum_{j=1}^{n} sim(z_i^{(0)}, z_i^{(1)})}, \quad (14)$$

where $\mathbb{1}(\cdot)$ denotes the indicator function.

As shown in Figure 4, the proposed approach substantially outperforms both SCCL and RSTC. These results demonstrate that the representations learned by the proposed method form a feature space with high intra-cluster compactness and clear inter-cluster separability.

### 4.4. Ablation Study

This section presents three groups of ablation experiments, covering all key components in the proposed model.

**Effectiveness of semantic consistency in CAOT.** As formulated in Eq. (3), the semantic consistency term $\langle S, QQ^T \rangle$ incorporates both the learnable attention similarity $S^{att}$ and the static cosine similarity $S^{cos}$. To evaluate the contribution of each component, we conduct ablation studies by selectively removing these similarities, i.e. w/o Cos, w/o

Att and w/o All. The results are shown in Table 3. The comparison between **OURS** and **w/o All** shows that incorporating sample-to-sample semantic consistency into the OT formulation increases accuracy significantly. The comparison between **OURS** with **w/o Att** and **w/o Cos** shows that the proposed method outperforms those using a single similarity. This is because exploring similarities in different spaces can capture authentic relationships between samples. Finally, the comparison between **w/o Att** and **w/o Cos** indicates that the proposed attention mechanism is more effective than the current method in capturing semantic relationships.

**Effectiveness of contrastive learning**. As detailed in Section 3.3, contrastive learning is first used to enhance the discriminability of sample representations, which serves as a prerequisite for similarity modeling. To evaluate its contribution, we perform ablation experiments by removing the contrastive learning component, i.e., **w/o CL**. As shown in Table 3, removing contrastive learning leads to a substantial performance degradation. These results validate that discriminative representations are essential for capturing the semantic similarity between samples, as they provided the necessary clarity to distinguish between samples.

*Table 3.* Ablation study results. We evaluate the contribution of each components by removing it from the framework.

| | Method | SearchSnippets | | StackOverflow | | GoogleNews-TS | |
|---|---|---|---|---|---|---|---|
| | | ACC | NMI | ACC | NMI | ACC | NMI |
| | w/o Cos | 78.81 | 66.14 | 83.52 | 73.89 | 81.83 | 92.68 |
| Ablation study 1 | w/o Att | 76.86 | 65.93 | 79.11 | 70.89 | 80.34 | 91.03 |
| | w/o All | 78.93 | 66.21 | 80.31 | 72.45 | 81.27 | 91.18 |
| Ablation study 2 | w/o CL | 56.65 | 30.66 | 80.51 | 71.74 | 75.12 | 90.26 |
| Ablation study 3 | w/o PL | 79.13 | 67.32 | 61.89 | 57.17 | 77.77 | 92.27 |
| Complete model | **OURS** | 80.12 | 67.40 | 85.96 | 75.43 | 83.53 | 93.15 |

*Table 4.* The Coefficient of Variation results on eight benchmark datasets, where the dataset names are shown in abbreviated form.

| | Agn | Sta | Bio | Sea | TS | T | S | Twe |
|---|---|---|---|---|---|---|---|---|
| CV | 0.03 | 0.17 | 0.18 | 0.31 | 0.60 | 0.52 | 0.54 | 0.61 |

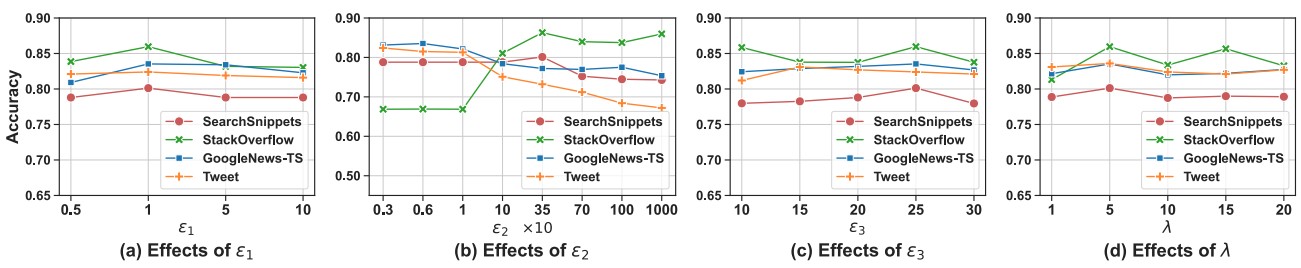

*Figure 5.* The effect of $\varepsilon_1$, $\varepsilon_2$, $\varepsilon_3$, and $\lambda$ on clustering accuracy.

**Effectiveness of Pseudo-Label Optimization**. We investigate the effectiveness of pseudo-labels for optimizing cluster assignments by removing the $L_P$ loss function, i.e., **w/o PL**. In this variant, the K-means algorithm is used for clustering instead. As shown in Table 3, the performance drops significantly without the $L_P$ loss. This demonstrates that the pseudo-labels generated by CAOT can effectively guide the clustering process and yield a more accurate partition than static clustering algorithms.

## 4.5. Hyperparameter Analysis

This section presents experiments to validate the effects of hyperparameters $\varepsilon_1$, $\varepsilon_2$, $\varepsilon_3$ and $\lambda$. Candidate sets for these parameters are $\{0.5, 1, 5, 10\}$ for $\varepsilon_1$, $\{0.03, 0.06, 0.1, 1, 3.5, 7, 10, 100\}$ for $\varepsilon_2$, $\{10, 15, 20, 25, 30\}$ for $\varepsilon_3$, and $\{1, 5, 10, 15, 20\}$ for $\lambda$. Experiments are conducted on the StackOverflow, SearchSnippets, GoogleNews-T, and Tweet datasets, which correspond to a balanced dataset, a slightly imbalanced dataset, an imbalanced dataset, and a severely imbalanced dataset, respectively.

The results are presented in Figure 5. As illustrated in Figures 5(a), (c), and (d), the proposed model exhibits robust performance across a wide range of values for $\varepsilon_1$, $\varepsilon_3$, and $\lambda$, demonstrating that the model is insensitive to these hyperparameters. The hyperparameter $\varepsilon_2$ governs the strength of the

uniformity constraint in Eq. (3), and it needs to be adjusted according to the specific imbalance level of dataset. Fortunately, Figure 5(b) shows that $\varepsilon_2$ allows a flexible adjustment range, i.e., the model remains stable when $\varepsilon_2$ ranges from 3.5 to 100 on balanced dataset (StackOverflow), from 0.03 to 3.5 on slightly imbalanced dataset (SearchSnippets), and from 0.03 to 0.1 on imbalanced dataset (GoogleNews-TS) and severely imbalanced dataset (Tweet). Therefore, a approximate estimate of the degree of category imbalance is sufficient to determine a suitable value of $\varepsilon_2$.

Building on this observation, we introduce an estimation method based on the Coefficient of Variation (CV). Specifically, we first use bge-large-en-v1.5 to obtain text embeddings, and then compute the CV value of the K-means clustering results based on these embeddings. The results are shown in Table 4. It can be seen that CV values exhibit significant variation across datasets with different imbalance levels. Based on this, we establish a step-wise mapping function that consider datasets with CV values in the ranges of $[0, 0.2)$, $[0.2, 0.4)$, $[0.4, 0.6)$, and $[0.6, 1.0]$ to be balanced, slightly imbalanced, imbalanced, and severely imbalanced, respectively. This protocol allows the proposed model to adaptively configure $\varepsilon_2$ for unseen datasets. Overall, the experiments demonstrate that the proposed model achieves adaptive hyperparameter configuration without manual hyperparameter tuning.

# 5. Cross-Domain Generalization

This section extends the proposed method to long-text datasets and image datasets to verify its cross-domain generalization capability.

**Long-text clustering.** We validate the generalizability of the proposed model in the long-text clustering using four datasets: the SyskillWebert (Pazzani, 1999), 20Newsgroups (Mitchell, 1997), MN-DS (Petukhova & Fachada, 2023), and Reuters (Lewis, 1997). We compare the model with TCLLM (Petukhova et al., 2025), a recent SOTA method for long-text clustering. The performance comparison is shown in Table 5. With the same encoder, the proposed model achieves better clustering performance than TCLLM. This demonstrates that the proposed approach generalizes well to long-document and exhibits cross-domain applicability.

**Image clustering.** In image clustering, we compare the proposed model with three methods: CC (Li et al., 2021), ICC-SPC (Guo et al., 2023), and discDC (Cai et al., 2025). The experiments are conducted on the STL-10 (Coates et al., 2011), CIFAR-10 (Krizhevsky et al., 2009) and CIFAR-100 (Krizhevsky et al., 2009) datasets. To ensure a fair comparison, we follow the feature extraction pipeline used in discDC. The result is reported in Table 6. It can be observed that the proposed method outperforms CC and ICC-SPC, and achieves performance comparable to discDC (a recent SOTA method specifically tailored for image clustering). These results show that our method generalizes well to image clustering tasks and exhibits cross-domain adaptability.

# 6. Related Work

**Pseudo-labeling based on optimal transport.** Optimal transport is a mathematical theory concerned with finding a transport matrix from one distribution to another while minimizing the total transport cost (Peyré et al., 2019; Laclau et al., 2017; Liu et al., 2023). Compared with greedy pseudo-labeling methods, OT-based methods consider the overall structure between sample distributions and cluster distributions to generate pseudo-labels (Qin et al., 2026; Caron et al., 2020; Cui et al., 2025), rather than assigning pseudo-labels to each sample in isolation (Berthelot et al., 2020; Del Barrio et al., 2019). SELA (Asano et al., 2020) is a representative work that uses OT to generate pseudo-labels by enforcing a uniform distribution over clusters. However, this method is not suitable for imbalanced datasets. PPOT (Zhang et al., 2024) and RSTC (Zheng et al., 2023) introduce a regularizer based on the imbalance level of predicted cluster probabilities, enabling them to handle imbalanced datasets. However, all these methods only consider relationships between samples and clusters, but ignore the inherent consistency between samples, which may assign different pseudo-labels to semantically similar samples.

*Table 5.* Clustering accuracies of long-text datasets.

| Method | SW | 20News | MN-DS | Reuters |
|---|---|---|---|---|
| TCLLM-Bert | 74.13 | 43.25 | 43.06 | 51.58 |
| **OURS** | 83.24 | 49.96 | 43.55 | 54.32 |

*Table 6.* Clustering accuracies in image clustering.

| Method | STL-10 | CIFAR-10 | CIFAR-100 |
|---|---|---|---|
| CC | 85.00 | 79.00 | 42.90 |
| ICC-SPC | 80.60 | 83.20 | 45.70 |
| discDC | 90.18 | 85.28 | 47.93 |
| **OURS** | 89.41 | 85.96 | 47.86 |

**Short Text Clustering.** Previous research can be divided into three categories: conventional methods, deep embedding methods, and deep joint clustering methods. Conventional methods rely on text statistical methods, such as BOW and TF-IDF, to extract features (Chowdhury, 2010). However, the representations produced by these methods are typically sparse and lack semantic information (Hadifar et al., 2019; Gao et al., 2021). Deep embedding methods use neural networks to extract features, after which clustering methods, such as K-means, are applied (Xu et al., 2017; Yong et al., 2024). However, feature extraction and clustering are often decoupled in these methods, which makes extracted representations unsuitable for clustering (Yao et al., 2025). Deep joint clustering methods integrate representation learning and clustering into a unified framework, allowing representation learning to be guided by the clustering objective (Xie et al., 2016; Zhang et al., 2021; Zheng et al., 2023; Yao et al., 2025).

# 7. Conclusion

In this paper, we propose an end-to-end short text clustering framework. Its effectiveness mainly stems from the proposed CAOT, which addresses the limitation of existing OT methods that ignore sample-to-sample semantic consistency. Extensive experiments on eight benchmark datasets demonstrate that the proposed framework outperforms state-of-the-art methods by a large margin. Beyond short texts clustering, the proposed framework can be generalized to long texts and image clustering tasks. Experiments also demonstrate that the proposed framework is robust to hyperparameter variations and has computational advantages over existing methods. The results suggest that the proposed framework can be applied to real-world clustering tasks.

# Impact Statement

This paper presents work whose goal is to advance the field of machine learning. There are many potential societal consequences of our work, none of which we feel must be specifically highlighted here.

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

## A. The Solution to the Optimization Problem in CAOT

### A.1. Formulation of the Solution

As mentioned in Section 3.2, the optimization problem in the proposed CAOT is:

$$\min_{\boldsymbol{Q},\boldsymbol{b}} \langle \boldsymbol{Q}, -\log(\boldsymbol{P}^{(0)})\rangle + \varepsilon_1 H(\boldsymbol{Q}) + \varepsilon_2 \Psi(\boldsymbol{b})^T \mathbf{1}_{k\times 1} - \varepsilon_3 \langle \boldsymbol{S}, \boldsymbol{Q}\boldsymbol{Q}^T\rangle \tag{15}$$

$$\text{s.t.} \, \boldsymbol{Q}\mathbf{1}_{k\times 1} = \boldsymbol{a}, \, \boldsymbol{Q}^T \mathbf{1}_{n\times 1} = \boldsymbol{b}, \, \boldsymbol{b}^T \mathbf{1}_{k\times 1} = 1,$$

where $\langle \cdot, \cdot \rangle$ is the Frobenius inner product; $\varepsilon_1$, $\varepsilon_2$ and $\varepsilon_3$ are weighting coefficients; $\boldsymbol{a} = \frac{1}{N}\mathbf{1}_{n\times 1}$; $H(\boldsymbol{Q}) = \langle \boldsymbol{Q}, \log(\boldsymbol{Q}) - 1\rangle$, and $\Psi(\boldsymbol{b}) = -\log(\boldsymbol{b}) - \log(\mathbf{1}_{k\times 1} - \boldsymbol{b})$.

To solve the problem, we propose an iterative method that integrates the Taylor expansion and the Lagrange multiplier. In the $t$-th ($t \geq 1$) iteration, the Taylor expansion of $\langle \boldsymbol{S}, \boldsymbol{Q}\boldsymbol{Q}^T\rangle$ truncated to the constant and linear terms is as follows:

$$\langle \boldsymbol{S}, \boldsymbol{Q}_{t-1}\boldsymbol{Q}_{t-1}^T\rangle + \langle (\boldsymbol{S}+\boldsymbol{S}^T)\boldsymbol{Q}_{t-1}, \boldsymbol{Q} - \boldsymbol{Q}_{t-1}\rangle, \tag{16}$$

in which $\frac{\partial \langle \boldsymbol{S},\boldsymbol{Q}\boldsymbol{Q}^T\rangle}{\partial \boldsymbol{Q}} = (\boldsymbol{S}+\boldsymbol{S}^T)\boldsymbol{Q}$ is used.

After substituting $\langle \boldsymbol{S}, \boldsymbol{Q}\boldsymbol{Q}^T\rangle$ in the objective function of Eq. (16) with its first-order Taylor approximation, one can get the following optimization problem:

$$\min_{\boldsymbol{Q},\boldsymbol{b}} \langle \boldsymbol{Q}, -\log(\boldsymbol{P}^{(0)})\rangle + \varepsilon_1 H(\boldsymbol{Q}) + \varepsilon_2 \Psi(\boldsymbol{b})^T \mathbf{1}_{k\times 1} - \varepsilon_3 \langle \boldsymbol{S}, \boldsymbol{Q}_{t-1}\boldsymbol{Q}_{t-1}^T\rangle - \varepsilon_3 \langle (\boldsymbol{S}+\boldsymbol{S}^T)\boldsymbol{Q}_{t-1}, \boldsymbol{Q}-\boldsymbol{Q}_{t-1}\rangle \tag{17}$$

$$\text{s.t.} \, \boldsymbol{Q}\mathbf{1}_{k\times 1} = \boldsymbol{a}, \boldsymbol{Q}^T\mathbf{1}_{n\times 1} = \boldsymbol{b}, \boldsymbol{b}^T\mathbf{1}_{k\times 1} = 1.$$

In the $t$-th iteration, with $\boldsymbol{Q}_{t-1}$ available, the objective function in Eq. (17) can be rewritten as follows:

$$
\begin{aligned}
& \langle \boldsymbol{Q}, -\log(\boldsymbol{P}^{(0)})\rangle + \varepsilon_1 H(\boldsymbol{Q}) + \varepsilon_2 \Psi(\boldsymbol{b})^T\mathbf{1}_{k\times 1} - \varepsilon_3\langle \boldsymbol{S}, \boldsymbol{Q}_{t-1}\boldsymbol{Q}_{t-1}^T\rangle - \varepsilon_3\langle (\boldsymbol{S}+\boldsymbol{S}^T)\boldsymbol{Q}_{t-1}, \boldsymbol{Q} - \boldsymbol{Q}_{t-1}\rangle \\
& = \langle \boldsymbol{Q}, -\log(\boldsymbol{P}^{(0)})\rangle - \varepsilon_3\langle (\boldsymbol{S}+\boldsymbol{S}^T)\boldsymbol{Q}_{t-1}, \boldsymbol{Q}\rangle + \varepsilon_1 H(\boldsymbol{Q}) + \varepsilon_2 \Psi(\boldsymbol{b})^T\mathbf{1}_{k\times 1} + C \\
& = \langle \boldsymbol{Q}, -\log(\boldsymbol{P}^{(0)}) - \varepsilon_3(\boldsymbol{S}+\boldsymbol{S}^T)\boldsymbol{Q}_{t-1}\rangle + \varepsilon_1 H(\boldsymbol{Q}) + \varepsilon_2 \Psi(\boldsymbol{b})^T\mathbf{1}_{k\times 1} + C,
\end{aligned} \tag{18}
$$

in which $C = \varepsilon_3\langle (\boldsymbol{S}+\boldsymbol{S}^T)\boldsymbol{Q}_{t-1}, \boldsymbol{Q}_{t-1}\rangle - \varepsilon_3\langle \boldsymbol{S}, \boldsymbol{Q}_{t-1}\boldsymbol{Q}_{t-1}^T\rangle$ is a constant. Therefore, the optimization problem in Eq. (17) can be rewritten as follows:

$$\min_{\boldsymbol{Q},\boldsymbol{b}} \, \langle \boldsymbol{Q}, \boldsymbol{M}\rangle + \varepsilon_1 H(\boldsymbol{Q}) + \varepsilon_2 \Psi(\boldsymbol{b})^T\mathbf{1}_{k\times 1} \tag{19}$$

$$\text{s.t.} \quad \boldsymbol{Q}\mathbf{1}_{k\times 1} = \boldsymbol{a}, \boldsymbol{Q}^T\mathbf{1}_{n\times 1} = \boldsymbol{b}, \boldsymbol{b}^T\mathbf{1}_{k\times 1} = 1,$$

with $\boldsymbol{M} = -\log(\boldsymbol{P}^{(0)}) - \varepsilon_3(\boldsymbol{S}+\boldsymbol{S}^T)\boldsymbol{Q}_{t-1}$.

The Lagrange multiplier method is used to solve the optimization problem in Eq. (19). The Lagrangian function is:

$$F(\boldsymbol{Q},\boldsymbol{b},\boldsymbol{\lambda},\boldsymbol{\tau},\nu) = \langle \boldsymbol{Q}, \boldsymbol{M}\rangle + \varepsilon_1 H(\boldsymbol{Q}) + \varepsilon_2 \Psi(\boldsymbol{b})^T\mathbf{1}_{k\times 1} - \boldsymbol{\lambda}^T(\boldsymbol{Q}\mathbf{1}_{k\times 1} - \boldsymbol{a}) - \boldsymbol{\tau}^T(\boldsymbol{Q}^T\mathbf{1}_{n\times 1} - \boldsymbol{b}) - \nu(\boldsymbol{b}^T\mathbf{1}_{k\times 1} - 1),$$

where $\boldsymbol{\lambda}$, $\boldsymbol{\tau}$ and $\nu$ are Lagrange multipliers. The partial derivative of $F(\boldsymbol{Q},\boldsymbol{b},\boldsymbol{\lambda},\boldsymbol{\tau},\nu)$ with respect to (w.r.t.) $Q_{ij}$ is:

$$\frac{\partial F(\boldsymbol{Q},\boldsymbol{b},\boldsymbol{\lambda},\boldsymbol{\tau},\nu)}{\partial Q_{ij}} = M_{ij} + \varepsilon_1 \log(Q_{ij}) - \lambda_i - \tau_j, \tag{20}$$

where $\lambda_i$ denotes the $i$-th element of $\boldsymbol{\lambda}$ and $\tau_j$ denotes the $j$-th element of $\boldsymbol{\tau}$. $M_{ij}$ and $Q_{ij}$ are the $(i,j)$-th elements of $\boldsymbol{M}$ and $\boldsymbol{Q}$, respectively. By setting $\frac{\partial F(\boldsymbol{Q},\boldsymbol{b},\boldsymbol{\lambda},\boldsymbol{\tau},\nu)}{\partial Q_{ij}} = 0$, one can get:

$$Q_{ij} = \exp\left(\frac{\lambda_i + \tau_j - M_{ij}}{\varepsilon_1}\right). \tag{21}$$

Due to the equality $\boldsymbol{Q}\mathbf{1}_{k\times 1} = \boldsymbol{a}$, one can get:

$$\sum_{j=1}^{k} Q_{ij} = \sum_{j=1}^{k} \exp\left(\frac{\lambda_i + \tau_j - M_{ij}}{\varepsilon_1}\right) = \exp\left(\frac{\lambda_i}{\varepsilon_1}\right)\sum_{j=1}^{k}\exp\left(\frac{\tau_j - M_{ij}}{\varepsilon_1}\right) = a_i, \tag{22}$$

where $k$ is the number of categories in the dataset. It is non-trivial to obtain:

$$\lambda_i = \varepsilon_1 \ln a_i - \varepsilon_1 \ln \sum_{j=1}^{k} \exp\left(\frac{\tau_j - M_{ij}}{\varepsilon_1}\right). \tag{23}$$

Similarly, due to the equality constraint $\boldsymbol{Q}^T \boldsymbol{1}_{k \times 1} = \boldsymbol{b}$, one can obtain:

$$\tau_j = \varepsilon_1 \ln b_j - \varepsilon_1 \ln \sum_{i=1}^{n} \exp\left(\frac{\lambda_i - M_{ij}}{\varepsilon_1}\right). \tag{24}$$

From Eq. (23) and Eq. (24), one can see that $\lambda_i$ and $\tau_j$ are functions of each other. When $\boldsymbol{b}$ is available, one can update $\lambda_i$ and $\tau_j$ in an iterative way.

Once $\boldsymbol{\lambda}$ and $\boldsymbol{\tau}$ are available, the next step is to get the value of $\boldsymbol{b}$. Specifically, the partial derivative of $F(\boldsymbol{Q}, \boldsymbol{b}, \boldsymbol{\lambda}, \boldsymbol{\tau}, \nu)$ in Eq. (20) w.r.t. $\boldsymbol{b}$ is:

$$\frac{\partial F(\boldsymbol{Q}, \boldsymbol{b}, \boldsymbol{\lambda}, \boldsymbol{\tau}, \nu)}{\partial b_j} = -\frac{\varepsilon_2}{b_j} + \frac{\varepsilon_2}{1 - b_j} + \tau_j - \nu, \tag{25}$$

where $b_j$ is the $j$-th element of $\boldsymbol{b}$. By setting $\frac{\partial F(\boldsymbol{Q}, \boldsymbol{b}, \boldsymbol{\lambda}, \boldsymbol{\tau}, \nu)}{\partial b_j} = 0$, one can get:

$$(\tau_j - \nu)b_j^2 - (\tau_j - \nu + 2\varepsilon_2)b_j + \varepsilon_2 = 0, \tag{26}$$

which is a second-order polynomial equation with the variable $b_j$. It is straightforward to verify that $\Delta_j = (\tau_j - \nu)^2 + 4\varepsilon_2^2 > 0$. Thus, there are two feasible solutions to Eq. (26), which are: $\frac{(\tau_j - \nu + 2\varepsilon_2) + \sqrt{\Delta_j}}{2(\tau_j - \nu)}$ and $\frac{(\tau_j - \nu + 2\varepsilon_2) - \sqrt{\Delta_j}}{2(\tau_j - \nu)}$, respectively. Since $\frac{(\tau_j - \nu + 2\varepsilon_2) + \sqrt{\Delta_j}}{2(\tau_j - \nu)} \geq 1$, it violates the domain constraint of $\Psi(\boldsymbol{b}) = -\log(\boldsymbol{b}) - \log(\boldsymbol{1}_{k \times 1} - \boldsymbol{b})$. Thus, $\frac{(\tau_j - \nu + 2\varepsilon_2) + \sqrt{\Delta_j}}{2(\tau_j - \nu)}$ is not a valid solution. In contrast, since $0 < \frac{(\tau_j - \nu + 2\varepsilon_2) - \sqrt{\Delta_j}}{2(\tau_j - \nu)} < 1$, it is a valid solution. Therefore,

$$b_j = \frac{(\tau_j - \nu + 2\varepsilon_2) - \sqrt{(\tau_j - \nu)^2 + 4\varepsilon_2^2}}{2(\tau_j - \nu)}. \tag{27}$$

Based on the constraint $\boldsymbol{b}^T \boldsymbol{1}_{k \times 1} = 1$, there is:

$$\sum_{j=1}^{k} \frac{(\tau_j - \nu + 2\varepsilon_2) - \sqrt{(\tau_j - \nu)^2 + 4\varepsilon_2^2}}{2(\tau_j - \nu)} = 1, \tag{28}$$

which is a function of $\nu$. One can use the Newton method to solve the equation in Eq. (28) to get the value of $\nu$. Once $\nu$ is available, $\boldsymbol{b}$ can be calculated by Eq. (27).

By iteratively updating $\{\boldsymbol{\lambda}, \boldsymbol{\tau}\}$ and $\{\nu, \boldsymbol{b}\}$, one can get the solution for the optimization problem in Eq. (19). The pseudo-code for solving the optimization problem in CAOT is presented in Algorithm 2.

### A.2. Convergence Verification of the Solution

Algorithm 2 presents an iterative algorithm for solving the optimization problem in the proposed CAOT. This section empirically verifies its convergence. Specifically, we evaluate the convergence behavior of CAOT by monitoring the pseudo-label change rate, defined as the fraction of CAOT-generated pseudo-labels that differ between consecutive iterations.

The results are presented in Figure 6. As shown in Figure 6(a), the change rate exhibits a sharp decline during the early training stages, indicating that the model rapidly learns the underlying cluster structure. As training progresses, the curve gradually flattens, reflecting the stabilization of the optimization process. Figure 6(b) provides a detailed view of the final iterations (1600–2000). One can observe that the change rate for most datasets consistently stabilizes below 4%, demonstrating that the proposed CAOT effectively reaches a steady and convergent state.

---

**Algorithm 2** The pseudo-code for solving the optimization problem in CAOT

---

1: **Input:** the probability matrix $\boldsymbol{P}^{(0)}$; the marginal constraint $\boldsymbol{a}$; the similarity matrix $\boldsymbol{S}$; weighting coefficients $\varepsilon_1$, $\varepsilon_2$ and $\varepsilon_3$.

2: **Output:** transport matrix $\boldsymbol{Q}$.

3: Initialize $\boldsymbol{b}$ from a uniform distribution so that $\boldsymbol{b}^T \mathbf{1}_{k \times 1} = 1$.

4: Initialize $\boldsymbol{Q}_0 = \boldsymbol{a}\boldsymbol{b}^T$.

5: **for** $t = 1$ to $T_1$ **do**

6:    $\boldsymbol{M} = -\log(\boldsymbol{P}^{(0)}) - \varepsilon_3(\boldsymbol{S} + \boldsymbol{S}^T)\boldsymbol{Q}_{t-1}$.

7:    The surrogate objective function of the optimization problem is $\langle \boldsymbol{Q}, \boldsymbol{M} \rangle + \varepsilon_1 H(\boldsymbol{Q}) + \varepsilon_2 \Psi(\boldsymbol{b})^T \mathbf{1}_{k \times 1}$.

8:    **for** $t_2 = 1$ to $T_2$ **do**

9:      With $\boldsymbol{b}$ fixed, update $\boldsymbol{\lambda}$ via Eq. (23) and $\boldsymbol{\tau}$ via Eq. (24).

10:      With $\boldsymbol{\lambda}$ and $\boldsymbol{\tau}$ fixed, use the Newton method to solve the equation in Eq.(28) to obtain $\nu$, and then update $\boldsymbol{b}$ via Eq. (27).

11:    **end for**

12:    Calculate $\boldsymbol{Q}_t$ via Eq. (21).

13: **end for**

14: $\boldsymbol{Q} = \boldsymbol{Q}_{T_1}$

---

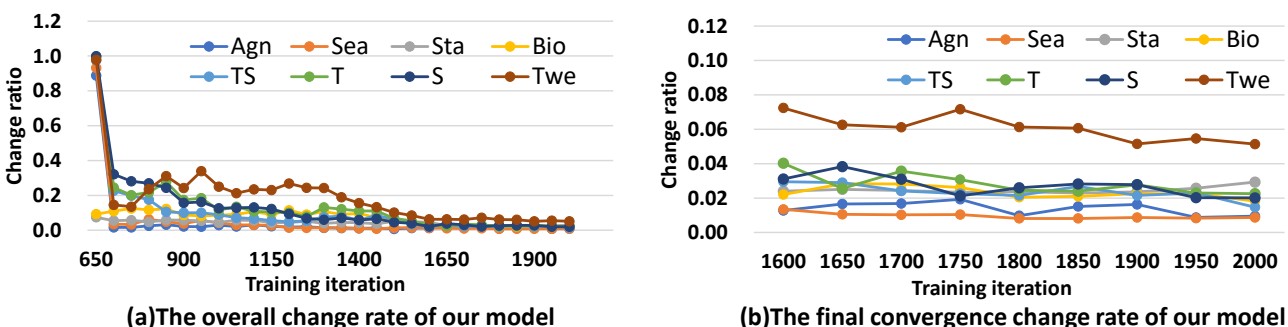

*Figure 6.* Convergence validation of the proposed method. The y-axis denotes the change rate of pseudo-labels.

## B. Supplementary Experiments

This section presents extensive supplementary experiments to further validate the effectiveness of the proposed model. §B.1 evaluates the effectiveness of CAOT, which is the core component that enables the proposed model to achieve SOTA performance. § B.2 examines the compatibility of the proposed framework with advanced encoders. §B.3 reports an objective comparison with existing methods in terms of computational budget. § B.4 investigates the reason why the model performs well on imbalanced datasets. §B.5 conducts comparative experiments on the feature space learned by the model.

### B.1. Comparison of Pseduo-labeling Effectiveness

To demonstrate the effectiveness of the proposed pseudo-labeling method (**CAOT-based PL**), we conduct comparative experiments against the prediction-based pseudo-labeling method (**Prediction-based PL**) and the AOT-based pseudo-labeling method (**AOT-based PL**). The AOT-based PL method is proposed in RSTC (Zheng et al., 2023), and remains the state-of-the-art OT method for short text clustering.

Experiments were conducted on the SearchSnippets, StackOverflow, and GoogleNews-TS datasets. We use ACC and NMI to evaluate the quality of pseudo-labels. The results are presented in Table 7. It can be observed that: (1) The pseudo-labels generated by **Prediction-based PL** are of low quality, as this method assigns pseudo-labels to samples independently. (2) **AOT-based PL** produces better pseudo-labels than the Prediction-based PL method by capturing the sample-to-cluster global structure. However, this method ignores sample-to-sample semantic consistency, resulting in suboptimal pseudo-labels. (3) **CAOT-based PL** generates reliable pseudo-labels by simultaneously considering both sample-to-sample semantic consistency and sample-to-cluster global structure.

*Table 7.* Results of different pseudo-labeling methods.

| Method | SearchSnippets | | StackOverflow | | GoogleNews-TS | |
|---|---|---|---|---|---|---|
| | ACC | NMI | ACC | NMI | ACC | NMI |
| Prediction-based PL | 72.43 | 59.42 | 69.18 | 63.54 | 69.18 | 63.54 |
| AOT-based PL | 76.81 | 63.82 | 79.63 | 69.78 | 78.34 | 88.88 |
| CAOT-based PL | 78.22 | 65.11 | 83.84 | 72.21 | 81.32 | 90.07 |

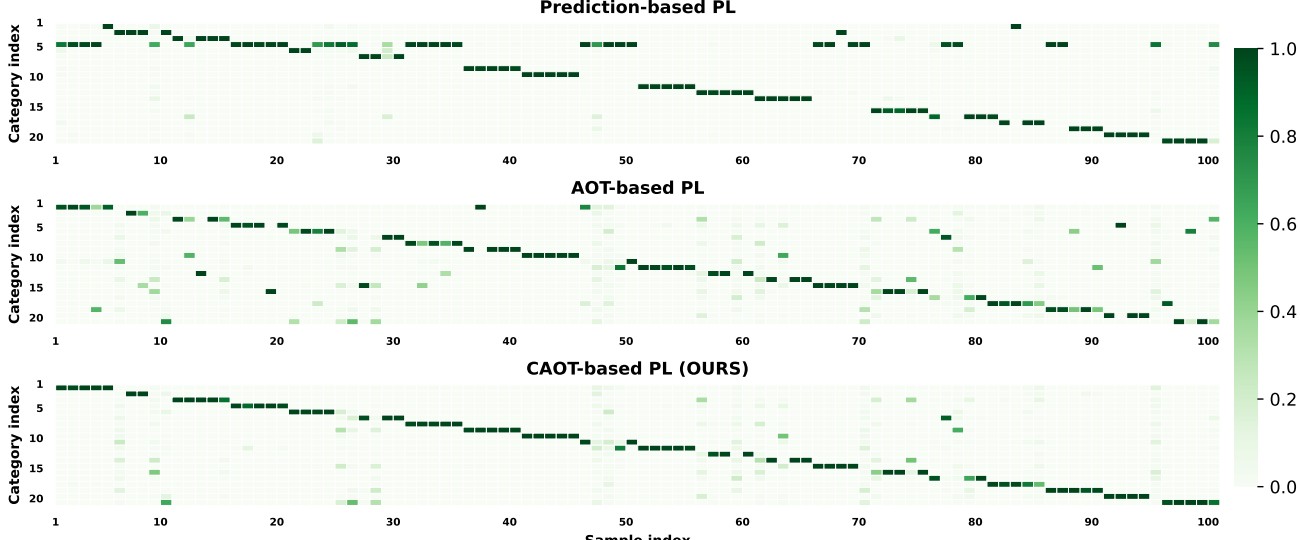

*Figure 7.* Visualization of the coupling matrix on StackOverflow. The x-axis represents the sample index, and the y-axis represents the category index. Each column represents the soft prediction of the corresponding sample. On the x-axis, every five consecutive samples belong to the same category.

Furthermore, to provide a detailed comparison among the various pseudo-labeling methods, we visualize their "coupling matrix" when generating pseudo-labels. In the Prediction-based PL experiment, the "coupling matrix" refers to the predicted probability assignment matrix. In the AOT-based PL and CAOT-based PL experiments, the "coupling matrix" refers to the transport matrix. We randomly selected five samples from each category in the StackOverflow datasets for visualization. Results are illustrated in Figure 7. One can observed that: (1) **Prediction-based PL** method produces incorrect soft-labels for some samples. (2) **AOT-based PL** achieves slightly better performance than Prediction-based PL. However, it exhibits a pronounced tendency to assign different pseudo-labels to semantically consistent samples, as evidenced by cases where samples belonging to the same category are frequently assigned different pseudo-labels. The reason for this issue is that existing OT methods ignore sample-to-sample semantic consistency. (3) **CAOT-based PL** method alleviates ambiguous predictions, and confidently assigns pseudo-labels to samples.

### B.2. Compatibility with Advanced Encoders

Experiments in this paper adopt SBERT as the backbone encoder, under which the proposed method already achieves state-of-the-art performance on most benchmark datasets. To further evaluate the generalizability of the proposed framework across different encoders, we conduct additional experiments by replacing SBERT with advanced text encoders, including bge-small-en-V1.5, bge-base-en-v1.5 (Xiao et al., 2023), and Qwen-3-Embedding (Zhang et al., 2025). The corresponding results are reported in Table 8.

As shown in Table 8, replacing SBERT with stronger encoders consistently improves the clustering accuracy of the proposed method across all datasets. In particular, compared with the SBERT-based variant, the Qwen3-Embedding-based model achieves substantial performance gains on several datasets, including StackOverflow, Biomedical, and Tweet. These improvements demonstrate that the proposed framework is not limited to a specific encoder, but can leverage advanced representation models.

*Table 8.* Clustering accuracies with different encoders.

| Method | Agn | Sea | Sta | Bio | TS | T | S | Twe |
|---|---|---|---|---|---|---|---|---|
| **OURS** (SBERT) | 87.30 | 80.12 | 85.96 | 47.37 | 83.53 | 73.47 | 79.57 | 82.36 |
| **OURS** (BGE-small) | 89.14 | 82.11 | 88.90 | 52.29 | 86.93 | 79.28 | 81.35 | 84.34 |
| **OURS** (BGE-base) | 90.21 | 86.33 | 90.45 | 53.17 | 87.53 | 82.36 | 83.42 | 86.04 |
| **OURS** (Qwen-3) | 91.06 | 83.28 | 92.38 | 54.33 | 90.19 | 85.32 | 86.81 | 90.60 |

*Table 9.* Comparison of model scale and training time between the proposed method and the baselines on different datasets. The units of "Model scale" and "Training time" are M and Min:Sec, respectively. Abbreviated names of datasets are used.

| Metric | Method | Agn | Sea | Sta | Bio | TS | T | S | Twe |
|---|---|---|---|---|---|---|---|---|---|
| | SCCL | 67.05 | 67.06 | 67.07 | 67.07 | 67.17 | 67.17 | 67.17 | 67.12 |
| Model size | RSTC | 68.24 | 68.24 | 68.25 | 68.25 | 68.35 | 68.35 | 68.35 | 68.30 |
| | **OURS** | 68.36 | 68.37 | 68.38 | 68.38 | 68.48 | 68.48 | 68.48 | 68.43 |
| | SCCL | 27:41 | 28:23 | 29:53 | 29:59 | 29:16 | 28:25 | 28:41 | 25:17 |
| Training time | RSTC | 15:47 | 8:59 | 23:16 | 23:47 | 21:41 | 20:13 | 21:56 | 12:41 |
| | **OURS** | 16:32 | 17:41 | 22:56 | 23:43 | 19:44 | 18:16 | 20:27 | 15:03 |

*Table 10.* Training time with different dataset size

| Method | 20,000 | 40,000 | 80,000 |
|---|---|---|---|
| RSTC | 23:16 | 38:56 | 1:01:12 |
| **OURS** | 22:56 | 25:20 | 30:53 |

It is worth noting that although the SBERT-based version already achieves highly competitive performance, the use of more powerful encoders further enhances the model performance substantially. This demonstrates the strong compatibility and scalability of the proposed method with advanced representation models, and further suggests its potential for future short text clustering scenarios where stronger pretrained encoders are available.

### B.3. Computation Budget

In this paper, all experiments are conducted on an NVIDIA GeForce RTX 3090 Ti GPU with the PyTorch framework. We compare the proposed model with SCCL and RSTC in terms of model size and training time. Detailed results are presented in Table 9. Three models are nearly the same in terms of model size, indicating that the proposed approach does not introduce additional computational overhead. In terms of training time, SCCL consumes more time than both RSTC and the proposed method. Moreover, compared with RSTC, the proposed method requires lower training time on large-scale datasets, including StackOverflow and Biomedical. The reason is that RSTC constructs and solves the optimal transport problem over the entire dataset in each iteration, whereas the proposed method operates on mini-batches. As a result, the per-iteration computation time of RSTC is longer than that of the proposed method.

To further evaluate how runtime scales with data size, we compare the proposed method with RSTC on expanded versions of the StackOverflow dataset. Specifically, we increase the dataset size from 20,000 samples to 40,000 and 80,000. As shown in Table 10, the runtime of RSTC rises significantly as the dataset size grows, revealing its limited scalability on large datasets. In contrast, the runtime of the proposed method remains stable regardless of dataset size. **These results demonstrate that the proposed method achieves superior computational efficiency and scalability, making it well suited for large-scale data clustering.**

### B.4. Analysis on Imbalanced Datasets

As shown in Table 2, the proposed model can be effectively applied to category-imbalanced datasets, including Search-Snippets, GoogleNews-TS, GoogleNews-T, GoogleNews-S, and Tweet. Given that contrastive learning can alleviate class imbalance to a certain extent, we conduct experiments to explore whether the model's adaptability to imbalanced datasets

*Table 11.* Comparison of clustering accuracies on imbalanced datasets. Abbreviated names of datasets are used.

|  | Agn | Sea | Sta | Bio | TS | T | S | Twe |
|---|---|---|---|---|---|---|---|---|
| Conventional OT | 87.32 | 73.89 | 85.83 | 46.98 | 62.03 | 58.42 | 61.52 | 47.09 |
| CAOT | 87.30 | 80.12 | 85.96 | 47.37 | 83.53 | 73.47 | 79.57 | 82.36 |

*Table 12.* Comparison of representation quality evaluated by K-means. Abbreviated names of datasets are used.

| Method | Agn | Sea | Sta | Bio | TS | T | S | Twe |
|---|---|---|---|---|---|---|---|---|
| KMeans (untrained) | 64.40 | 54.47 | 60.50 | 38.30 | 65.49 | 59.75 | 59.54 | 53.80 |
| KMeans (on RSTC) | 78.53 | 71.21 | 73.20 | 42.58 | 78.54 | 66.09 | 69.83 | 68.71 |
| KMeans (on OURS) | 82.10 | 77.06 | 78.24 | 44.05 | 78.81 | 66.93 | 71.47 | 72.42 |
| RSTC | 85.99 | 79.83 | 80.07 | 45.69 | 83.30 | 73.10 | 78.11 | 77.75 |
| OURS | 87.30 | 80.12 | 85.96 | 47.37 | 83.53 | 73.47 | 79.57 | 82.36 |

stems from CAOT or contrastive learning. In the experiment, CAOT is replaced by conventional OT while other components remain unchanged. The performance degradation on imbalanced datasets indicates that CAOT solves the class imbalance problem, while stable performance suggests that contrastive learning plays the decisive role. As shown in Table 11, clustering performance declines significantly when CAOT is removed, which confirms that the proposed model's adaptability to imbalanced datasets is primarily attributed to CAOT.

### B.5. Comparison of Representation Quality

In Section 4.3, we evaluate the learned representations via cosine similarity. This section provides a complementary analysis from another perspective. Since K-means clustering results reflect the intrinsic quality and separability of the learned feature space, we apply K-means clustering to quantitatively assess the representation quality.

In this experiment, RSTC is used as a comparison. As shown in Table 12, K-means clustering on the optimized representations outperforms that on untrained ones, demonstrating that both methods can learn high-quality representations. In particular, the results show that representations learned by the proposed method achieves better performance. It can also be observed that the K-means clustering is inferior to the original approach. This reveals that although applying K-means to high-quality representations achieves good clustering performance, developing advanced clustering algorithms still remains essential.

## C. Supplementary Experimental Setups

This section provides supplementary details of the experimental setup. §C.1 elaborates on the datasets used in this work. §C.2 provides detailed definitions of evaluation metrics, and §C.3 presents the main experimental configurations.

### C.1. Datasets

We conducted experiments on eight datasets: AgNews, StackOverflow, Biomedical, SearchSnippets, GoogleNews-TS, GoogleNews-T, GoogleNews-S and Tweet. Brief descriptions of these datasets are as follows:

- **AgNews** is extracted from the AG news corpus (Zhang et al., 2015), which contains 8,000 news headlines and is categorized into four topics (Rakib et al., 2020).

- **SearchSnippets** is extracted from web search transactions. It consists of 12,340 web search results and is divided into eight categories (Phan et al., 2008).

- **StackOverflow** consists of 20,000 question titles drawn from 20 technical categories (Xu et al., 2017). These samples are randomly selected from Kaggle competition data and include a mix of technical discussions and programming queries.

- **Biomedical** contains 20,000 research paper titles from 20 scientific topics (Xu et al., 2017). The data is sourced from BioASQ and reflects the typical professional vocabulary and structure of scientific literature.

- **GoogleNews** comprises 11,109 articles covering 152 events (Yin & Wang, 2016). The dataset is provided in three variants: full articles (**GoogleNews-TS**), titles only (**GoogleNews-T**), and snippets only (**GoogleNews-S**).

- **Tweet** contains 2,472 tweets linked to 89 distinct queries (Yin & Wang, 2016), collected from the Text Retrieval Conference's microblog tracks in 2011 and 2012, reflecting the informality of social media communication.

## C.2. Evaluation Metrics

Consistent with previous works (Rakib et al., 2020; Zheng et al., 2023), we employ two standard metrics to evaluate clustering performance: Accuracy (ACC) and Normalized Mutual Information (NMI). Accuracy measures the proportion of correct clustered texts, which is defined as follows:

$$ACC = \frac{\sum_{i=1}^{N} \mathbb{1}\left(y_i = \text{map}\left(\tilde{y}_i\right)\right)}{N}, \tag{29}$$

where $N$ is the dataset size, $y_i$ denotes the true label, and $\tilde{y}_i$ denotes the predicted label. $\mathbb{1}(\cdot)$ is the indicator function. $\text{map}(\cdot)$ is a function that employs the Hungarian algorithm to align predicted labels with true labels (Papadimitriou & Steiglitz, 1998).

Normalized Mutual Information measures the mutual dependence between ground-truth and predicted labels by normalizing mutual information with their entropy:

$$NMI(\boldsymbol{Y}, \tilde{\boldsymbol{Y}}) = \frac{I(\boldsymbol{Y}, \tilde{\boldsymbol{Y}})}{\sqrt{H(\boldsymbol{Y})H(\tilde{\boldsymbol{Y}})}}, \tag{30}$$

where $\boldsymbol{Y}$ denotes the set of true labels, $\tilde{\boldsymbol{Y}}$ denotes the set of predicted labels, $I(\cdot, \cdot)$ denotes mutual information, and $H(\cdot)$ represents entropy.

## C.3. Experiment Settings

The Clustering Network $G_p$ is an MLP network with dimensions 768-768-768-$k$, where the first two layers use ReLU activation function, and the last layer uses a softmax activation function. The Projecting Network $G_z$ is an MLP network with dimensions 768-768-128, where the first layer uses the ReLU activation function and, there are no activation functions in the second layer. The temperature parameters for attention loss and contrastive learning loss are $\tau_A = 1$ and $\tau_I = 1$, respectively. The number of iterations, including both $T_1$ and $T_2$ in the solution to CAOT, is set to 10. The batch size $n$ is 200. The total number of training iterations $E_{total}$ is 2,000. The number of warm-up iterations $E_{warm}$ is 600 for all datasets except Stackoverflow and Biomedical, where the warm-up stage is omitted. The hyperparameters $\varepsilon_1 = 1$, $\varepsilon_3 = 25$, and $\lambda = 5$ are fixed across all datasets, while $\varepsilon_2$ is set to 100, 3.5, 0.06, and 0.03 for balanced, slightly imbalanced, imbalanced, and severely imbalanced datasets, respectively. When employing data augmentation, the *contextual augmenter* (Kobayashi, 2018) is configured as in SCCL (Zhang et al., 2021) and RSTC (Zheng et al., 2023) to generate two augmented texts by substituting the top-n suitable words of the input text.

