# OpenReview forum: "Beyond Looking Up, Try Looking Around: Harmonizing Global Structure and Local Consistency in Optimal Transport for Short Text Clustering"
_ICML.cc/2026/Conference — ICML 2026 regular_

### Official Review · Reviewer_zoqt · 2026-02-28

**Soundness:** 3
**Presentation:** 4
**Significance:** 2
**Originality:** 2
**Overall Recommendation:** 4
**Confidence:** 4

**Summary:**

The paper proposes to integrate semantic consistency into existing OT-based short text clustering methods by an instance-level attention network. The semantic relationships between samples are then integrated into existing OT methods to achieve more accurate and robust short text clustering. Experiments show that the proposed CAOT outperforms state-of-the-art approaches.

**Compliance With Llm Reviewing Policy:**

Affirmed.

**Final Justification:**

My concerns are well-addressed.

**Key Questions For Authors:**

1. Is it possible to compare COAT with RSTC in more details, especially for compoents other than the attention network and $\epsilon_3<S,QQ^T>$ in Eq (3)? Are they exactly the same?
2. How efficient and scalable is COAT compared to other baselines, especially RSTC?
3. Is it possible to compare COAT with GSDMM on a few benchmarking datasets?
4. Is it possible to add some additional experiments by using modern LLM-based sentence encoders?

**Limitations:**

yes

**Strengths And Weaknesses:**

## Strengths

1. The motivation of integrating semantic consistency into OT-based method is clearly demonstrated by both Figure 1 and the ablation study.
2. The optimization of Eq (3), which is the core contribution of the paper, is mathematically rigorous and seems to be at the same complexity of the Sinkhorn algorithm for solving the original entropic OT problem, which is relevantly efficient.
3. COAT is applicable to other clustering tasks such as long-text and image clustering, and shows strong performances.
4. The paper is generally well-structured, clearly written, and easy to follow

## Weaknesses

1. It seems that COAT is based on a former work, RSTC, with an added attention network to better capture the semantic relationship at instance level. However, a detailed comparison with RSTC is lacking in the paper.
2. From Table 2, it seems that COAT only slightly outperforms RSTC on most datasets, but with an added attention network that requires training, which may negatively impact the efficiency and scalability of the model
3. The encoder adopted by this paper, SBERT, is outdated. It’s unclear whether semantic inconsistency will still occur with modern LLM-based sentence encoders, such as OpenAI text-embedding-3 and Qwen3-Embedding
4. Some recent and published baselines for short text clustering are missing in Table 2, e.g., GSDMM[1]
4. Other relatively minor weaknesses exist: 1) a formal definition of short text clustering is missing; 2) Eq (1) is the definition of entropic OT, not OT in general.

### References
[1] Cheng, Enhao, et al. "An Enhanced Model-based Approach for Short Text Clustering." arXiv preprint arXiv:2507.13793 (2025).

---

> ### Author Rebuttal · Authors · 2026-03-28
>
> We sincerely thank you for the time and effort you have devoted to reviewing our manuscript. Below, we provide point-by-point responses, with some similar issues addressed together.
>
> >**W1 and Q1:** Is it possible to compare COAT with RSTC in more details, especially for components other than the attention network and $\epsilon_3<S,QQ^T>$ in Eq (3)? Are they exactly the same?
>
> **Reply:**
> 1. **Our core contribution lies in revolutionizing the modeling ideology of discrete OT, filling the critical gap where existing OT commonly ignore local modeling. This limitation is verified in Figure 6 of the manuscript.** To this end, we propose a general solution paradigm, where $\epsilon_3 \langle S, QQ^T \rangle$ in Eq. (3) is the mathematical design to realize this paradigm. RSTC is merely the test subject we use to evaluate whether our paradigm is effective.
> 2. **By incorporating local structural modeling, COAT successfully scales OT execution to the mini-batch level. In contrast, RSTC must execute OT on the entire dataset, which severely limits its scalability on large-scale datasets. This phenomenon is confirmed by Table 9 in the manuscript.**
> 3. The proposed attention network is a novel instance-level module designed to capture the local relation between sentences during discrete OT alignment. It is fundamentally different from existing token-level attention.
>
> >**W2 and Q2:** How does COAT compare with other baselines, especially RSTC, in terms of efficiency and scalability?
>
> **Reply:** The reviewer’s insights are highly professional. **Since this question is identical to Reviewer RnqB’s Weakness 2, we respectfully and kindly invite** the reviewer to our detailed response there due to the rebuttal word limit.
>
> >**W3 and Q4:** The encoder adopted by this paper, SBERT, is outdated. Is it possible to add some additional experiments by using modern LLM-based sentence encoders? Such as OpenAI text-embedding-3 and Qwen3-Embedding.
>
> **Reply:** Following your suggestion, we evaluate our model with bge-small-en-v1.5，bge-base-en-v1.5 and Qwen3-Embedding as encoders. Since OpenAI text-embedding-3 cannot satisfy the gradient requirements of our framework, we replace it with the BGE models to ensure the robustness of results. **As shown in the table below, our framework naturally accommodates these latest encoders and achieves a substantial performance leap. We will include these results in the camera-ready version.**
>
> ||Agn|Sea|Sta|Bio|TS|T|S|Twe|
> |-|-|-|-|-|-|-|-|-|
> |Ours|87.3|80.1|86.0|47.4|83.5|73.5|79.6|82.4|
> |Ours(bge-small)|89.1|82.1|88.9|52.3|86.9|79.3|81.4|84.3|
> |Ours(bge-base)|90.2|86.3|90.5|53.2|87.5|82.4|83.4|86.0|
>
> >**W4 and Q3:** Some recent and published baselines for short text clustering are missing in Table 2, e.g., GSDMM. Is it possible to compare COAT with GSDMM on a few benchmark datasets?
>
> **Reply:**
> 1. **We add GSDMM in the table below and further include other recent methods proposed in 2024–2025.** The results show that our method consistently outperforms CLSESSP, FNSCC, and SCPCL. Although GSDMM, which is designed for multi-class datasets, performs better on TS, T, S, and Twe, it performs poorly on low-class datasets such as Agn, Sea, Sta, and Bio. our method maintains top-tier performance across all datasets. We will include these comparisons in the camera-ready stage.
> 2. **Our method is a general paradigm applicable across different domains, rather than one that pursues extreme gains on a few specific datasets.** As evidenced by Tables 4 and 5, our model also achieves SOTA on image and long-text tasks. In contrast, GSDMM does not exhibit applicability to other domains.
> ||Agn|Sea|Sta|Bio|TS|T|S|Twe|
> |-|-|-|-|-|-|-|-|-|
> |CLSESSP[1]|80.5|69.9|74.4|40.3|64.5|63.6|64.6|57.9|
> |FNSCC[2]|87.1|85.2|70.9|35.5|81.0|68.6|71.7|81.7|
> |SCPCL[3]|85.8|79.0|81.0|44.3|87.0|73.0|79.1|80.3|
> |GSDMM|21.5|43.2|24.4|34.1|87.4|81.1|83.1|85.5|
> |Ours|87.3|80.1|86.0|47.4|83.5|73.5|79.6|82.4|
> ```python
> [1]CLSESSP: Contrastive learning of sentence embedding with strong semantic prototypes, KBS 2024.
> [2]FNSCC: Fuzzy Neighborhood-Aware Self-Supervised Contrastive Clustering for Short Text, EMNLP 2025.
> [3]Constructing a robust Short-Text Clustering Model for contrastive learning based on optimized adaptive optimal transport for pseudo-label generation,EAAI 2025.
> ```
>
> >**W5:** Other relatively minor weaknesses exist: 1) a formal definition of short text clustering is missing; 2) Eq.(1) is the definition of entropic OT, not OT in general.
>
> **Reply:** We sincerely thank you for pointing out these details. We will make the following revisions at the camera-ready stage:
> 1. We will add the following definition in Line 43: “Short text clustering aims to partition short texts in an unsupervised manner, assigning semantically similar samples to the same cluster while separating semantically distinct samples into different clusters.”
> 2. We will revise the wording here to “entropic OT”

---

> > ### Author Rebuttal · Reviewer_zoqt · 2026-04-02
> >
> > Thank you for your detailed response, which partially addressed my concerns. My remaining concerns are
> >
> > **W3 and Q4: modern LLM-based sentence encoders**
> >
> > Thank you for the additional experimental results. My initial concerns are actually 1) whether modern LLM-based sentence encoders would affect the effectiveness of semantic consistency in CAOT (i.e., what would the ablation study in Table 3 in the paper look like if using modern  LLM-based sentence encoders), and 2) whether LLM-based encoders + other baselines (e.g., RSTC) would outperform LLM-based sentence encoders + CAOT
> >
> > **Performance comparison with GSDMM**
> >
> > From the table, it seems that CAOT is outperformed by GSDMM on imbalanced datasets (TS, T, S, and Twe). The authors claim that CAOT handles imbalanced datasets well (Table 11), but comparison with GSDMM seems contradictory to this conclusion.

---

> > > ### Author Response · Authors · 2026-04-03
> > >
> > > We sincerely admire the reviewer’s professionalism and conscientiousness. We are truly grateful to the reviewer for continuing to dedicate substantial time and effort to the careful review of our manuscript. Below, we provide point-by-point responses to the questions raised.
> > >
> > >
> > > > **Question 1.1: Performance Comparison of Different Methods Using Qwen3-Embedding.**
> > >
> > > **Reply:** We report the comparison of Qwen3-Embedding + different methods. The results show that, after switching to an LLM-based encoder, the final performance of both CAOT and the baselines improves. Importantly, CAOT achieves the largest improvement.
> > >
> > > The reason is that replacing an LLM-based encoder improves the quality of sample representations, which naturally enhances the performance of different methods. Beyond this natural improvement, CAOT can model inter-sample similarity more accurately on top of these high-quality representations, which helps CAOT better enforce semantic consistency and thus brings larger performance gains.
> > >
> > > |Method|Agn|Sea|Sta|Bio|TS|T|S|Twe|
> > > |-|-|-|-|-|-|-|-|-|
> > > |RSTC (Qwen3)|87.34|**84.21**|86.77|47.44|85.72|77.30|81.37|84.42|
> > > |CLSESSP (Qwen3)|82.40|73.46|77.83|43.48|72.64|69.05|71.75|78.37|
> > > |FNSCC (Qwen3)|87.54|80.18|83.17|44.61|84.11|76.51|82.63|83.61|
> > > |SCPCL (Qwen3)|87.37|83.52|84.16|48.94|87.10|79.78|80.53|**91.36**|
> > > |Ours (Qwen-3)|**91.06**|83.28|**92.38**|**54.33**|**90.19**|**85.32**|**86.81**|90.60|
> > >
> > > PS: The above comparison does not include GSDMM+, because it is a word-frequency-based method and is not applicable to embedding extraction.
> > >
> > > > **Question 1.2: Ablation Study After Switching the Encoder.**
> > >
> > > **Reply:** We further report the ablation results of CAOT + Qwen3-Embedding. The “w/o All” result indicates that a large portion of the improvement brought by the encoder switch stems from semantic consistency. The reason is that high-quality representations lead to a more accurate $S$ matrix, which in turn makes $\langle S, QQ^T \rangle$ more effective. The results of "w/o Cos" and "w/o Att" show that exploring similarities in different spaces can capture a more accurate $S$ matrix. The conclusion drawn from this ablation experiment is consistent with that of the manuscript.
> > >
> > > PS: The only notable difference after switching encoders is that, "w/o Att" underperforms "w/o All" with SBERT, whereas "w/o Att" outperforms "w/o All" with Qwen3. This is because the representation quality when using SBERT is relatively low, and cosine similarity alone cannot produce an accurate $S$ matrix. Conversely, with Qwen3, using only cosine similarity can also enable $S$ to play a positive role.
> > >
> > > |Method|Sea-ACC|Sea-NMI|Sta-ACC|Sta-NMI|TS-ACC|TS-NMI|
> > > |-|-|-|-|-|-|-|
> > > |w/o Cos (SBERT)|78.81|66.14|83.52|73.89|81.83|92.68|
> > > |w/o Att (SBERT)|76.86|65.93|79.11|70.89|80.34|91.03|
> > > |w/o All (SBERT)|78.93|66.21|80.31|72.45|81.27|91.18|
> > > |Ours (SBERT)|80.12|67.40|85.96|75.43|83.53|93.15|
> > > ||||||||
> > > |w/o Cos (Qwen3)|82.06|70.33|90.46|78.32|87.51|94.65|
> > > |w/o Att (Qwen3)|80.51|68.42|85.87|75.94|85.05|92.97|
> > > |w/o All (Qwen3)|79.05|67.51|82.46|74.38|83.45|92.26|
> > > |Ours (Qwen3)|83.28|70.86|92.38|80.52|90.19|95.16|
> > >
> > >
> > > >**Question2: Performance comparison with GSDMM.**
> > >
> > > **Reply:** We would like to respectfully clarify the context of this claim. The statement of "CAOT can effectively handle imbalanced datasets" was drawn based on the comparative experiments in the original manuscript.
> > >
> > > We greatly appreciate you reminding us about the recently emerged work, GSDMM+, in the research community. We added a comprehensive comparison with GSDMM+ in our first-round rebuttal. As you correctly noted, GSDMM+ slightly outperforms CAOT on imbalanced datasets. However, this advantage comes at a severe cost to generalizability. Specifically, it suffers notable performance drops on balanced datasets with fewer categories (e.g., Agn, Sea, Sta, Bio). In contrast, CAOT exhibits superior robustness, maintaining highly competitive and stable performance across diverse data distributions.
> > >
> > > In the camera-ready version, we will revise our claims regarding “performance on imbalanced datasets” by taking GSDMM+ into consideration. The specific revisions are as follows: 1. Formally include GSDMM+ in the baseline comparison tables of the manuscript. 2. Add the following discussion in Section 4.2.
> > > ```python
> > > "The results show that GSDMM+ performs slightly better than CAOT on imbalanced datasets. However, this advantage of GSDMM+ comes at the cost of sacrificing the model’s generalization ability. When applied to balanced datasets (e.g., Agn, Sea, Sta, and Bio), the performance of GSDMM+ drops significantly. In contrast, CAOT maintains highly competitive and stable performance across different data distributions."
> > > ```
> > >
> > > We once again commit to incorporating the corresponding revisions into the manuscript in the camera-ready version.
> > >
> > > ———————————————————————————————————————————
> > >
> > > **Update: Thank you for recognizing our response and for raising the score from 3 to 4.**

---

### Official Review · Reviewer_3mv2 · 2026-03-09

**Soundness:** 3
**Presentation:** 3
**Significance:** 2
**Originality:** 2
**Overall Recommendation:** 4
**Confidence:** 5

**Summary:**

The paper appears to study the concept of improving pseudo-label generation for short text clustering by integrating sample-to-sample semantic consistency into the optimal transport (OT) framework. Specifically, the authors propose a Consistency-Aware Optimal Transport (CAOT) formulation that jointly considers global sample-to-cluster structure and local semantic similarity through an instance-level attention mechanism and contrastive representation learning.

**Compliance With Llm Reviewing Policy:**

Affirmed.

**Final Justification:**

The paper appears to study the concept of improving pseudo-label generation for short text clustering by integrating sample-to-sample semantic consistency into the optimal transport (OT) framework. My concerns are well addressed by rebuttal. Therefore, I improve my score to weak accept.

**Key Questions For Authors:**

- The method relies on pseudo-labels generated during training, which may propagate errors when initial representations are weak or when clusters are highly overlapping.


- The attention-based semantic similarity matrix introduces additional computational complexity that may limit scalability for very large datasets.

- The comparative experiments rely on relatively outdated baselines, with most methods proposed before 2023. It remains unclear how the effectiveness of the proposed method can be convincingly demonstrated when the comparisons are primarily conducted against older approaches.

- The conceptual novelty of the method appears limited. Since the framework mainly combines existing components such as contrastive learning, attention mechanisms, and OT-based pseudo-labeling, could the authors clarify what the key methodological innovation is beyond this integration?
﻿
- The performance improvements over strong baselines are modest on several datasets. Could the authors elaborate on whether these gains are statistically significant and explain the practical advantages of the proposed modifications compared to existing approaches?

**Limitations:**

Please see the weaknesses and questions.

**Strengths And Weaknesses:**

Strengths:

- The paper identifies a limitation of existing OT-based pseudo-labeling methods, namely the lack of modeling semantic consistency among neighboring samples.


- The framework integrates several components—contrastive representation learning, attention-based similarity modeling, and optimal transport pseudo-labeling—into a unified EM-style training pipeline.


- Experimental evaluation covers multiple short-text datasets to analyze the contribution of proposed method.



Weaknesses:


- The conceptual novelty of the method appears limited, as the framework mainly combines existing components (contrastive learning, attention mechanisms, and OT-based pseudo-labeling) rather than introducing fundamentally new learning principles.


- The performance improvements over strong baselines are modest on several datasets, raising questions about the practical significance of the proposed modifications.

---

> ### Author Rebuttal · Authors · 2026-03-28
>
> We sincerely thank the time and effort you have devoted to reviewing our manuscript.
>
> >**Reply for Q1 and Q2:** The reviewer’s insights are highly professional. **Since Question 1 and Question 2 align perfectly with the concerns raised in Reviewer RnqB’s Weakness 1 and Weakness 2, respectively, we respectfully and kindly invite the reviewer to our response there** due to the rebuttal word limit.
>
> >**Q3:** The comparative experiments rely on relatively outdated baselines, with most methods proposed before 2023.
>
> **Reply:** Thank you for your suggestion. We add comparisons with four recent methods proposed in 2024–2025. **The results show that our method consistently outperforms CLSESSP, FNSCC, and SCPCL.**  The latest GSDMM+ is specifically designed for multi-class datasets and performs better than ours on TS, T, S, and Twe. However, it performs very poorly on low-class datasets, i.e., Agn, Sea, Sta, and Bio. In contrast, our method maintains top-tier performance across all types of datasets. **We will include these comparisons in the camera-ready version.**
>
> ||Agn|Sea|Sta|Bio|TS|T|S|Twe|
> |-|-|-|-|-|-|-|-|-|
> |CLSESSP[1]|80.5|69.9|74.4|40.3|64.5|63.6|64.6|57.9|
> |FNSCC[2]|87.1|85.2|70.9|35.5|81.0|68.6|71.7|81.7|
> |SCPCL[3]|85.8|79.0|81.0|44.3|87.0|73.0|79.1|80.3|
> |GSDMM+[4]|21.5|43.2|24.4|34.1|87.4|81.1|83.1|85.5|
> |Ours|87.3|80.1|86.0|47.4|83.5|73.5|79.6|82.4|
>
> ```python
> [1]CLSESSP: Contrastive learning of sentence embedding with strong semantic prototypes, KBS 2024.
> [2]FNSCC: Fuzzy Neighborhood-Aware Self-Supervised Contrastive Clustering for Short Text, EMNLP 2025.
> [3]Constructing a robust Short-Text Clustering Model for contrastive learning based on optimized adaptive optimal transport for pseudo-label generation,EAAI 2025.
> [4]An Enhanced Model-based Approach for Short Text Clustering, arxiv 2025.
> ```
>
> >**W1 and Q4:** The framework mainly combines existing components such as contrastive learning, attention mechanisms, and OT-based pseudo-labeling, could the authors clarify what the key methodological innovation is beyond this integration?
>
> **Reply:** We want to clarify that our key innovation lies not in inventing a new network, but in proposing a novel theoretical paradigm to address a fundamental bottleneck in OT.
> 1. Our motivation is based on a critical observation: existing discrete OT focuses only on global distribution alignment while overlooking local modeling, which often leads to severe mismatching errors. **This is a shared limitation, which is verified in Figure 6 of the manuscript.**
> 2. **Our proposition is to explicitly integrate local structural modeling into OT to compensate this shared limitation.**  Table 6 and Figure 6 provide strong support for our claim. This is a innovation of the OT ideology. The proposed OT paradigm (CAOT) is not limited to short-text scenarios, but is also applicable to any other discrete OT scenarios.
> 3. **Contrastive learning and the attention network are merely tools for local structural modeling. The resulting local similarity is then incorporated in CAOT, while the formulation of CAOT itself is the core contribution of our work.**  Lines 201–211 clearly provide the rationale for introducing this tools. We also conducted sufficient ablation studies to verify the contribution of each component:  w/o Cos, w/o Att, and w/o All for the attention network; w/o CL for contrastive learning.
>
> >**W2 and Q5:** The performance improvements over strong baselines are modest on several datasets. Could the authors elaborate on whether these gains are statistically significant and explain the practical advantages of the proposed modifications compared to existing approaches?
>
> **Reply:**
> 1. **Our model is a general framework that achieves top-tier performance across different domains, rather than extreme gains on only a few specific datasets.** As evidenced by Tables 4 and 5, our model also achieves SOTA on image and long-text tasks. Compared with baselines that rely on extensive hyperparameter tuning to fit a single dataset (Appendix B.2 demonstrates that our hyperparameters are adaptive across different datasets), it is reasonable that our paradigm brings moderate improvements on some specific short-text datasets. All of current baselines does not exhibit applicability to other domains.
> 2. **The performance gains of our model are closely related to dataset characteristics and are more pronounced on datasets that rely on local modeling.**  For example, StackOverflow dataset contains heavily overlapping programming-language vocabulary (some real cases are provided below), where our method achieves a 5.89% improvement. On datasets less dependent on local modeling, our method yields moderate and stable gains.
> ```python
> # "sql query" recurs across different cases.
> label 2: Issue in Oracle Sql Query
> label 5:VBA SQL Query Table Error problem
> label 18:Linq to Sql query using "not in"
> label 15: testing an sql query in php
> Label 17:file path from sql query in drupal
> ```

---

> > ### Author Rebuttal · Reviewer_3mv2 · 2026-04-06
> >
> > Thanks for the author's response. After reading the rebuttal, my concerns are addressed. Therefore, I will improve my score.

---

> > > ### Author Response · Authors · 2026-04-07
> > >
> > > We are deeply encouraged to know that our rebuttal and additional analyses have adequately addressed the reviewer’s concerns. We sincerely thank the reviewer for the careful attention devoted to our response, as well as for the continued time and effort invested in evaluating our work.

---

### Official Review · Reviewer_RnqB · 2026-03-10

**Soundness:** 2
**Presentation:** 2
**Significance:** 2
**Originality:** 2
**Overall Recommendation:** 4
**Confidence:** 3

**Summary:**

This paper proposes Consistency-Aware Adaptive Optimal Transport (CAOT) for short text clustering, addressing a key limitation in existing OT-based pseudo-labeling methods that overlook sample-to-sample semantic consistency. CAOT integrates both sample-to-cluster global structure and sample-to-sample semantic relationships through a novel OT formulation with learnable attention-based similarity. The framework jointly optimizes pseudo-label generation, semantic similarity construction, and clustering via an EM procedure. Extensive experiments on eight benchmarks demonstrate significant improvements over state-of-the-art methods, with additional validation on long-text and image datasets showing strong cross-domain generalization.

**Compliance With Llm Reviewing Policy:**

Affirmed.

**Final Justification:**

The responses adequately address most of my concerns.

**Key Questions For Authors:**

Refer to the weaknesses.

**Limitations:**

Refer to the weaknesses.

**Strengths And Weaknesses:**

Strengths

1. The paper identifies a meaningful limitation in existing OT-based pseudo-labeling method, namely, their neglect of sample-level semantic consistency. The proposed CAOT formulation is a principled extension that integrates both global and local structures, which is well-motivated.

2. The CAOT formulation is mathematically rigorous, with a clearly defined objective function and a carefully derived optimization algorithm based on the Majorization-Minimization framework. The inclusion of both cosine similarity and learnable attention-based similarity adds flexibility and expressiveness.

3. The proposed method was evaluated on eight publicly available datasets, including StackOverflow, Biomedical, among others. The improvements in both ACC and NMI are substantial, surpassing not only traditional baselines such as K-means and SCCL but also recent state-of-the-art models based on optimal transport. Furthermore, ablation studies were conducted to examine the impact of the attention module, the optimal transport module, and various hyperparameter settings on the final clustering performance, thereby underscoring the necessity of each individual component.

Weaknesses & Questions

1. The efficacy of the CAOT framework is intrinsically tied to the fidelity of the nearest neighbor search within the initial embedding space. While the attention matrix Satt is dynamically updated, the process remains a form of self-bootstrapping that is susceptible to a "cold-start" problem: if initial embeddings fail to capture the true topological manifold, the neighborhood-aware OT formulation may inadvertently reinforce early misassignments, a phenomenon akin to confirmation bias, where the model increasingly trusts and propagates its initially flawed neighbor relationships. To improve robustness, the authors are encouraged to discuss or evaluate mitigation strategies, such as an annealing schedule for the consistency penalty or the integration of uncertainty estimation to weight the reliability of initial neighborhood structures.

2.The "looking around" mechanism necessitates computing a similarity matrix between samples. The instance-level attention network typically exhibits a computational complexity of O(N^2). When applied to massive-scale short text corpora, such as millions of tweets, this quadratic complexity could impose prohibitive computational overhead and memory consumption. The paper's discussion on scalability to extremely large datasets is somewhat limited.

3.The evaluation is primarily conducted on standard text datasets. It would be beneficial to assess the model's performance on highly colloquial and non-standardized text, such as social media data rich in abbreviations and emojis, as well as in multilingual settings. Such experiments would provide valuable insights into the stability and robustness of the local consistency modeling under extreme noise conditions.

---

> ### Author Rebuttal · Authors · 2026-03-28
>
> We sincerely thank the reviewer for the time and comment, and below we provide point-by-point responses.
>
> >**W1:** Would introducing local similarity between samples into CAOT cause a 'cold-start' issue? To improve robustness, the authors are encouraged to discuss or evaluate mitigation strategies, such as an annealing schedule or the integration of uncertainty estimation.
>
> **Reply:** The “cold-start” is indeed a challenge of EM-based methods. The reviewer’s insight of this key point is highly professional. **We had already taken this issue into consideration and designed a dedicated mechanism for it in our manuscript**
> 1. **Our manuscript included a dedicated Warm-up mechanism to address the “initial quality” issue (Lines 250–255, Algorithm 1).** Specifically, the model first performed a warm-up using only contrastive learning, which improved representation discriminability and laid a solid foundation for subsequent similarity modeling, thereby alleviating the blind “initial quality” concern you raised.
> 2. **The ablation study (w/o CL) in Table 3 showed that the model indeed suffered from the “cold-start” without Warm-up, whereas Warm-up effectively mitigated this issue.**
> 3. **Our framework can flexibly incorporate the latest encoder models, fundamentally addressing your concern by improving the quality of the initial embedding space.** Specifically, we evaluate bge-small-en-v1.5 and bge-base-en-v1.5 as encoders. The results in the table below show that our framework naturally accommodates these encoders, yields substantial performance gains, and even reduces training cost through faster convergence.
>
> ||Agn|Sea|Sta|Bio|TS|T|S|Twe|
> |-|-|-|-|-|-|-|-|-|
> |Ours-ACC|87.3|80.1|86.0|47.4|83.5|73.5|79.6|82.4|
> |Ours(bge-small)-ACC|89.1|82.1|88.9|52.3|86.9|79.3|81.4|84.3|
> |Ours(bge-base)-ACC|90.2|86.3|90.5|53.2|87.5|82.4|83.4|86.0|
> |Ours-Time|16:32|17:41|22:56|23:43|19:44|18:16|20:27|15:03|
> |Ours(bge-small)-Time|11:37|14:51|22:56|24:07|14:26|14:54|14:21|12:10|
> |Ours(bge-base)-Time|12:37|13:11|24:53|24:40|15:05|14:04|14:26|12:07|
>
> >**W2:** The "looking around" mechanism necessitates computing a similarity matrix between samples. The instance-level attention network typically exhibits a computational complexity of $O(N^2)$. On massive-scale datasets, this quadratic complexity may lead to huge costs. The paper's discussion on scalability to extremely large datasets is somewhat limited.
>
> **Reply:**
> 1. **The proposed attention network is executed within mini-batches, and therefore the computational complexity introduced by it is $O(\frac{N}{B}× B^2)$ rather than $O(N^2)$**，where N is the dataset size and B is the batch size. Since O(B2) is fixed, increasing the data only leads to a $O(\frac{N}{B})$ linear increase in complexity, rather than an exponential one. Compared with the computation of Encoder, the$O(\frac{N}{B}× B^2)$ is negligible.
> 2. **Appendix B.3 and Table 9 of the manuscript provided a detailed analysis of scalability on large-scale datasets.** The results showed that our training time remained stable as the data scale increased (Sample size 20,000→80,000; Training time 22:56→30:53), whereas that of the current SOTA method, RSTC, increased significantly (Sample size 20,000→80,000; Training time 23:16→1:01:12).
>
> >**W3:** The evaluation mainly focuses on standard text datasets. It would be beneficial to assess the model on highly colloquial and non-standardized text, such as social media data, as well as in multilingual settings. Such experiments would provide valuable insights into the necessity of local modeling under extreme conditions.
>
> **Reply:**
> 1. **The datasets used in this paper strictly follow previous works [1-4], and that are widely adopted standard benchmarks in the short text clustering field.** To ensure fair comparisons, all evaluations are conducted on exactly the same data. Therefore, the current experimental setup is comprehensive.
> 2. The insight regarding testing robustness on social media data is highly perceptive. **The "Tweet" dataset used in manuscript is exactly the Twitter comments, which contain severe colloquialisms and multiple languages.** Several real cases are provided below to illustrate its noise. Despite these highly noisy, our method improves accuracy by up to 4.61% over the current SOTA method, demonstrating the necessity of local modeling under extreme noise.
> ```python
> Case1: “bruce will fave kardashian lol” (colloquial + abbreviation)
> Case2: “omg oprah plege wouldn text drive didn seriously lol xd bb” (abbreviation + non-standard text)
> Case3: “loooooooooooove ncis” (character elongation)
> Case4:“journalist van trouw mishandeld ro nieuws”(multilingual: Dutch)
> ```
> ```python
> [1] Contrastive Learning Subspace for Text Clustering, Arxiv 2024.
> [2] Supporting clustering with contrastive learning, NAACL 2021.
> [3] Self-taught convolutional neural networks for short text clustering, Neural Networks 2017.
> [4] A self-training approach for short text clustering, RepL4NLP 2019.
> ```

---

> > ### Author Rebuttal · Reviewer_RnqB · 2026-04-04
> >
> > I thank the authors for their detailed rebuttal. The responses adequately address most of my concerns. I will rasie my score.

---

> > > ### Author Response · Authors · 2026-04-04
> > >
> > > We are very pleased to learn that our clarifications and additional analyses have adequately addressed the reviewer’s concerns. We once again thank the reviewer for the recognition of our efforts and for the time and care devoted to evaluating our work.

---

### Official Review · Reviewer_bGgn · 2026-03-13

**Soundness:** 4
**Presentation:** 4
**Significance:** 3
**Originality:** 3
**Overall Recommendation:** 5
**Confidence:** 4

**Summary:**

This paper proposes CAOT, a clustering framework that improves pseudo-label generation in OT-based short text clustering. Existing OT approaches mainly focus on the relationship between samples and cluster prototypes, which can result in semantically similar samples receiving different pseudo-labels. Such inconsistencies may propagate during iterative training and degrade representation quality.
To address this issue, the authors introduce a semantic-consistency aware OT formulation. The method first learns semantic relationships between samples through an instance-level attention mechanism. These relationships are then incorporated into the OT objective so that the transport process considers both global cluster structure and local semantic consistency. The resulting pseudo-labels are used to iteratively refine the representation model. The proposed method is evaluated on multiple short text clustering benchmarks and consistently outperforms existing baselines. The authors further show that the approach can generalize beyond short text by applying it to long text and image clustering tasks.

**Compliance With Llm Reviewing Policy:**

Affirmed.

**Key Questions For Authors:**

How sensitive is the method to inaccuracies in the attention-based estimation of semantic relationships between samples?
Could the authors provide additional ablation studies isolating the contribution of each component of the proposed framework?
How does the computational complexity of CAOT compare with standard OT-based clustering approaches when scaling to larger datasets?

**Limitations:**

yes

**Strengths And Weaknesses:**

[Strengths]

Soundness
The paper addresses a well defined limitation of existing OT based clustering methods. By incorporating semantic relationships between samples into the transport process, the method directly targets a known issue in pseudo-label generation. The proposed formulation is technically sound and integrates naturally with existing OT based clustering frameworks.
The learning pipeline is coherent: the attention module estimates semantic relationships, the OT solver generates pseudo-labels incorporating this information, and the representation model is updated using these pseudo-labels. This iterative training scheme is logically consistent and aligns well with prior pseudo-labeling approaches.
Empirically, the paper presents results on several short text datasets and demonstrates consistent improvements over competitive baselines. The additional experiments on long text and image clustering also suggest that the idea is not limited to a specific data modality.

Presentation
The paper is generally well written and clearly organized. The motivation is straightforward, and the method is introduced in a step-by-step manner that makes it easy to follow the overall pipeline. The figures illustrating the proposed framework help clarify how semantic relationships are integrated into the OT formulation.

Significance
Clustering short text data remains challenging due to sparse and noisy representations. Improving pseudo-label quality is therefore an important research direction. The proposed approach provides a practical way to incorporate semantic consistency into clustering and may be useful for a range of representation learning tasks.

While the work is primarily focused on clustering, the idea of integrating semantic consistency into OT based learning frameworks could potentially be applied in other contexts.

Originality
The novelty lies in augmenting the OT based pseudo-labeling process with sample level semantic consistency information. While the approach builds on existing techniques such as attention-based similarity modeling and OT clustering, the integration of these ideas into a unified framework is well motivated and leads to measurable empirical gains.

[Weaknesses]
Incremental methodological novelty
Although the method is well motivated, the core contribution can be interpreted as an extension of existing OT based clustering frameworks with an additional semantic consistency term. The idea is intuitive and useful, but it may be perceived as incremental rather than a fundamentally new approach to clustering.


Limited robustness analysis
The method relies on the attention module to estimate semantic relationships between samples. If these estimates are noisy or unreliable, the consistency constraint could potentially propagate incorrect information through the clustering process. The paper would benefit from additional analysis examining the robustness of the method under such conditions.


Component-level evaluation
While the empirical results demonstrate improvements over baselines, the paper could provide more detailed ablation studies to clarify the contributions of individual components of the framework. In particular, isolating the impact of the attention based similarity modeling versus the OT modification would help better understand the source of the observed performance gains.

---

> ### Author Rebuttal · Authors · 2026-03-28
>
> We sincerely thank the reviewer for the recognition of our manuscript, as well as for the time and effort devoted to reviewing it. Below, we provide point-by-point responses to address the Weaknesses and Questions.
>
> >**W1:** Although the method is well motivated, the core contribution can be interpreted as an extension of existing OT based clustering frameworks with an additional semantic consistency term. The idea is intuitive and useful, but it may be perceived as incremental rather than a fundamentally new approach to clustering.
>
> **Reply:**
> 1. We would like to clarify that our contribution is not merely the proposal of a new method. More importantly, we identify and analyze the lack of local modeling as a common limitation in this line of OT-based approaches. **This shared limitation is verified in Figure 6 (AOT-based PL) of the manuscript. Our work propose a generalized paradigm to address this shared limitation.** This paradigm is not limited to short-text domain, but can also be extended to various discrete OT applications.
>
> 2. **Table 6 and Figure 6 (CAOT-based PL) demonstrate that our paradigm can effectively address this shared limitation. Meanwhile, Tables 4 and 5 demonstrate that our paradigm is also applicable to image and long-text tasks.**
>
> >**W2 and Q1:** The method relies on the attention module to estimate semantic similarity between samples. If these estimates are noisy, the consistency constraint may propagate incorrect information during clustering.The paper would benefit from additional analysis examining the robustness of the method under such conditions.
>
> **Reply:** The reviewer’s insight of this key point is highly professional. If similarity is estimated directly in the initial low-quality feature space, the issue of “noisy estimates” does indeed arise. We had already taken this issue into consideration and designed a dedicated mechanism for it during the algorithm design stage.
>
> 1. **Our manuscript included a dedicated Warm-up mechanism to address the “noisy estimates” issue (Lines 250–255, Algorithm 1).** During model training, the model did not directly perform similarity modeling and OT optimization, but instead only executed the contrastive learning to Warm-up model. This mechanism was specifically designed to improve the discriminability of the sample representations and laid a foundation for subsequent similarity estimating, thereby avoiding the blind “estimates noisy” concern you raised .
> 2. **The ablation study (w/o CL) in Table 3 showed that the model indeed suffered from “noisy estimates” without Warm-up, whereas Warm-up effectively mitigated this issue.**
>
>
>
> >**W3 and Q2:** While the empirical results demonstrate improvements over baselines, the paper could provide more detailed ablation studies to clarify the contributions of individual components of the framework. In particular, isolating the impact of the attention based similarity modeling versus the OT modification would help better understand the source of performance gains.
>
> **Reply: We would like to clarify that the ablation study in Table 3 basically contains all the components of our model.** Specifically, our framework consists of three sub-models in total: (a) PGM, (b) SSCM, and (c) SGM. Ablation Study 1, 2, and 3 in Table 3 correspond to the ablation of these three sub-models, respectively. Further, in Ablation Study 1, we conducted three further ablations. These were: completely eliminating the local semantic modeling in CAOT (w/o All), eliminating the similarity based on attention (w/o Att), and eliminating the attention based on cosine similarity (w/o Cos). Among them, w/o Att is exactly the setting you referred to for isolating the attention modeling from the OT modification.
>
>
> >**Q3:** How does the computational complexity of CAOT compare with standard OT-based clustering approaches when scaling to larger datasets?
>
> **Reply:**
> 1. **The proposed attention network is executed within mini-batches, and therefore the computational complexity introduced by it is $O(\frac{N}{B} * B^2)$**，where N is the dataset size and B is the batch size. Since $O(B^2)$ is fixed, increasing the data only leads to a $O(\frac{N}{B})$ linear increase in complexity, rather than an exponential one. Compared with the computation of Encoder, the $O(\frac{N}{B} * B^2)$ is negligible.
> 2. **Appendix B.3 and Table 9 of the manuscript provided a detailed analysis of scalability on large-scale datasets.** The results showed that our training time remained stable as the data scale increased (Sample size 20,000 → 80,000; Training time 22:56 → 30:53), whereas that of the current SOTA method, RSTC, increased significantly (Sample size 20,000 → 80,000; Training time 23:16 → 1:01:12).

---

> > ### Author Rebuttal · Reviewer_bGgn · 2026-04-04
> >
> > Thank you for the clear and helpful rebuttal. The authors have adequately addressed my concerns, particularly regarding robustness (mechanism and ablation), component contributions, and scalability. I have no further concerns.

---

> > > ### Author Response · Authors · 2026-04-04
> > >
> > > We sincerely thank the reviewer for carefully reading our rebuttal and for providing a positive assessment. We are glad that our clarifications and additional analyses have adequately addressed the reviewer’s concerns.

---

### Decision · Program_Chairs · 2026-04-30

**Decision:**

Accept (regular)

**Comment:**

This paper addresses a clear limitation of optimal-transport-based short text clustering by incorporating local semantic consistency into pseudo-label generation. Reviewers broadly agreed that the motivation is sound, the formulation is technically careful, and the optimization procedure is well developed. The empirical evaluation across multiple datasets was viewed positively, and reviewers appreciated that the method appears applicable beyond a single benchmark setting.

The main concerns were about the degree of novelty and the practical effect size. Several reviewers viewed the work as a principled integration of known ingredients rather than a fundamentally new clustering paradigm, and they asked for a stronger comparison to closely related OT-based methods as well as clearer discussion of why the improvements are meaningful when gains are modest on some datasets.

The rebuttal was effective: concerns about positioning and empirical support were largely addressed, and one reviewer explicitly raised their score after rebuttal. Overall, the work is technically solid, well evaluated, and likely to be useful to the community even if its novelty is more incremental than radical. I therefore recommend acceptance.